# DeBLAS: Accelerate LLM Pretraining by Length-based Sequence Scheduling

## Abstract

Pretraining large language models (LLMs) is computationally intensive, typically requiring massive datasets and training iterations. Although recent advances in data selection have shown improvement in training efficiency, their gains often diminish under scaling laws. In this work, we dive into the impact of sequence length on language model pretraining and propose a length-based online data scheduling method for acceleration. Specifically, we design a dense-balanced sequence scheduling framework for LLM pretraining: 1) at the first stage, the model is exposed to uniform-length dense token batches to encourage the formation of global language representations; 2) the second stage incorporates variable-length sequences, which reinforces learned abstractions while significantly reducing the total number of training iterations. Our idealized theoretical analysis suggests that our approach can accelerate language model pretraining through the gradient variance reduction achieved by the combination of both stages. Empirical results show that our approach achieves comparable perplexity to standard pretraining with substantially fewer optimization steps, pinpointing a promising way to reduce the computational burden of LLM pretraining.

## 1 Introduction

Scaling of model parameters and training data has led to substantial improvements in the Large Language Model (LLM) capabilities (Brown et al., 2020; Chowdhery et al., 2023; Touvron et al., 2023), yet this progress comes at the cost of enormous computational overhead for auto-regressive pretraining (Rae et al., 2021; Hoffmann et al., 2022; Achiam et al., 2023). The time-consuming phase, which often lasts hundreds of thousands of GPU days, poses critical challenges in terms of development cycles and resource demands (Liu et al., 2024; Yang et al., 2024). In parallel with advances in hardware (Fan et al., 2025) or optimization (Zhao et al., 2024a; Zhang et al., 2024b) aspects, recent explorations in data selection (Xie et al., 2023a; Wettig et al., 2024; Tirumala et al., 2023; Wang et al., 2024; Pouransari et al., 2024) have shown promising results in improving training efficiency, revealing a significant potential for acceleration.

However, existing data selection methods have a substantial gap from being adapted to LLM pretraining. Offline data selection (Xie et al., 2023b; Wettig et al., 2024; Ye et al., 2024) usually requires an additional reference model (Mindermann et al., 2022; Deng et al., 2023) that is trained either on held-out data or publicly available benchmarks, which may be constrained in real-world scenarios. In contrast, online data selection (Mindermann et al., 2022; Hong et al., 2024; Nguyen et al., 2024; Wang et al., 2024) provides a cost-effective strategy without additional model requests, as it is generally adopted in post-training. Nevertheless, current online data selection methods primarily depend on model-centric criteria such as perplexity (Jiang et al., 2019) or gradient dynamics (Nguyen et al., 2024; Wang et al., 2024) and often fail to explicitly consider foundational textual attributes as a dimension for curating training data.

In this work, we investigate the data scheduling of LLM pretraining from the perspective of sequence length, which plays an important role in acceleration (refer to Figure 1). The sequence length in natural language not only represents the complexity of semantic meanings but also affects the pretraining data processing. We systematically explore the learning dynamics of varying the sequence length and identify one key observation. On the one hand, dense batches, composed of uniform-length semantically continuous sequences, enable the

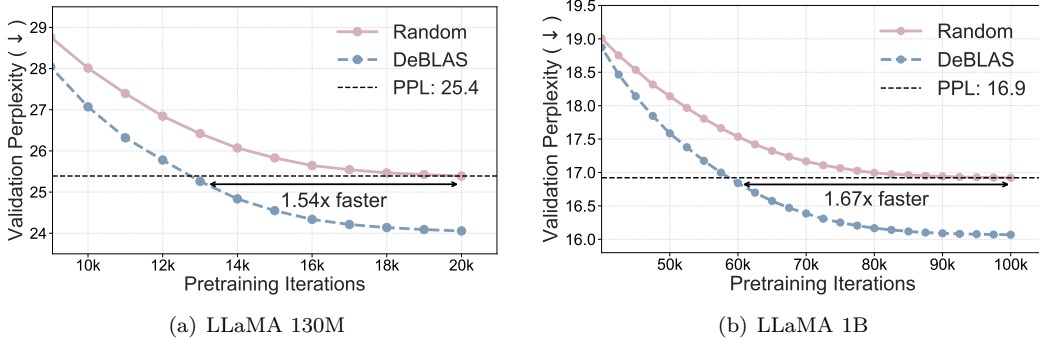

Figure 1: **Performance comparison** in pretraining acceleration evaluated by validation perplexity on the C4 dataset between random batch selection and the proposed Dense-Balanced Sequence Scheduling (DeBLAS). DeBLAS achieves the target validation perplexity (PPL) 1.54x faster on LLaMA 130M and 1.67x faster on LLaMA 1B in terms of required iterations.

model to achieve lower or comparable perplexity than variable-length padded batches. On the other hand, prolonged exposure to such uniform-length data induces length-wise biases, requiring extra calibration effort.

Motivated by the above findings, we propose a two-stage online data scheduling method, namely, Dense-Balanced Sequence Scheduling (DeBLAS), to accelerate LLM pretraining: as illustrated in the right of Figure 2, at the *Dense Batching* stage, the model is trained exclusively on dense batches to maximize the effective utilization of the token batch size budget. At the *Balanced Batching* stage, we include sequences of diverse lengths in batches to rectify the length bias induced by the Dense Batching stage. Specifically, we randomly hold out a small subset of the training corpus as a calibration set and divide its samples into different length bins. During the training, we periodically evaluate the model on the calibration set and update the sampling probability of each length bin based on its evaluation loss and proportion in the calibration set. The high token utilization efficiency achieved during the first phase enables the model to more efficiently optimize over the latter variable-length sequences, while the second phase counteracts the length bias of the initial stage through dynamic sampling. To sum up, our main contributions are:

- Technically, we focus on sequence length as an effective dimension for acceleration, and design a novel online data scheduling (termed as DeBLAS) that leverages dense-to-balanced sequence length progression to accelerate LLM pretraining (in Sections 3.1 and 3.2).

- Theoretically, we provide an idealized analysis to show that, under simplifying assumptions, DeBLAS can accelerate the convergence of language model pretraining through the gradient variance reduction achieved by the combination of its Dense Batching and Balanced Batching stage (in Section 3.3).

- Empirically, we conduct extensive experiments to verify the effectiveness of DeBLAS on two widely used pretraining corpora, i.e., C4 and SlimPajama, and perform various ablation studies and further discussions to provide a thorough understanding (in Section 4).

## 2 Preliminaries

**Problem setup.** We consider pretraining language model $\theta$ on the task of next token prediction using the cross-entropy loss function. Let $\mathcal{D}$ denote the distribution of training corpus where the sequence $\boldsymbol{x} = (x_1, x_2, \ldots, x_{|\boldsymbol{x}|})$ is sampled i.i.d. from $\mathcal{D}$. The vanilla training objective is defined as $\min_\theta \mathcal{L}(\theta) = \mathbb{E}_{\boldsymbol{x} \sim \mathcal{D}} \left[ - \sum_{i=1}^{|\boldsymbol{x}|} \log P_\theta(x_i \mid x_{<i}) \right]$, where $x_{<i}$ denotes the sub-sequence of $\boldsymbol{x}$ before the $i^{\text{th}}$ position. We consider the problem of pretraining acceleration under a fixed token batch size $N_B$, which is widely adopted in LLM pretraining (Brown et al., 2020). At each iteration, we can access a text sequence mini-batch $B = \{\boldsymbol{x}^j\}_{j=1}^{B_{\text{S}}}$, where $B_{\text{S}}$ denotes the number of sequences in the batch. All sequences in $B$ are tokenized and preprocessed to a uniform length $L_B$. The token batch size $N_B = B_{\text{S}} L_B$ represents the total count of tokens in the mini-batch $B$, including special tokens such as padding tokens and end-of-sequence (EOS)

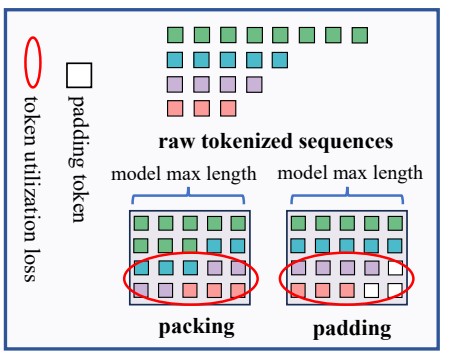 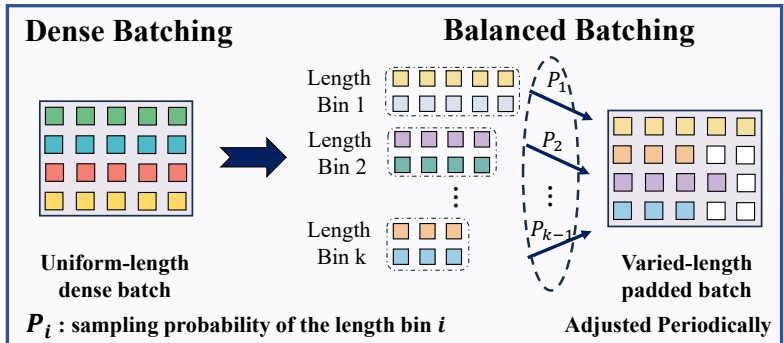

Figure 2: **Left:** Packing reduces token utilization when splitting full sequences into disjoint segments. Padding reduces token utilization due to semantically meaningless padding tokens. **Right:** The pretraining process consists of two stages: (*Dense Batching*) The model is trained on uniform-length dense batches for maximum token utilization. (*Balanced Batching*) The model is trained on variable-length padded batches to rectify the length bias from the Dense Batching stage, with the sampling probabilities for different length bins adjusted periodically based on the current model state.

tokens, which directly determines the computational demands of every training iteration. Our objective is to minimize the number of training iterations required for model $\theta$ to reach a given performance score by dynamically selecting data to construct the mini-batch $B$ in every iteration.

**Data preprocessing.** Large-scale text corpora contain massive variable-length text sequences. *Padding* and *packing* are two common strategies to curate them into structured-sized matrices or tensors. As illustrated in the left panel of Figure 2, the padding strategy extends shorter sequences with padding tokens to a fixed length $L$, while longer sequences are truncated to $L$ (Zhao et al., 2024a; Han et al., 2024). Instead, the packing strategy concatenates multiple sequences together and then splits them into chunks of the fixed length $L$ (Brown et al., 2020; Pagliardini et al., 2023).

**Token utilization rate.** Inspired by Pouransari et al. (2024), we define a concept, the token utilization rate (TUR), to quantitatively measure the overall utilization efficiency of tokens in a batch during autoregressive pretraining. Let $L_B$ denote the uniform sequence length of the batch $B$, and $A(x_i^j)$ denote the number of semantically meaningful tokens which the token $x_i^j$, the $i^{\text{th}}$ token of sequence $\boldsymbol{x}^j$, can attend to. The TUR is defined as follows:

$$\text{TUR} = \frac{\sum_{j=1}^{B_{\text{S}}} \sum_{i=1}^{|\boldsymbol{x}^j|} A(x_i^j)}{B_{\text{S}} L_B}. \tag{1}$$

Padding tokens cause damage to the TUR according to Eq. (1), whereas packing also encounters significant loss of TUR when complete documents are split into different segments and segments of different documents are packed into the same sequence. Empirically, in Appendix A.3.1, we observe that packing may require more optimization iterations than padding for the model to reach the same validation perplexity under a fixed token batch size. Furthermore, packing obscures the influence of the length of every individual sequence on the TUR and training dynamics. We thus employ padding for data preprocessing in all subsequent explorations and experiments. We leave a complete discussion of related works in Appendix B.

## 3 Methodology

### 3.1 Motivation and Systematic Exploration

Current online data selection methods for language models (Katharopoulos & Fleuret, 2018; Hong et al., 2024; Wang et al., 2024) predominantly rely on model-related metrics such as perplexity scores or gradients, while neglecting fundamental textual characteristics. At the same time, some previous works progressively increase the sequence lengths according to predefined length curricula during pretraining to improve optimization efficiency (Jin et al., 2023; Li et al., 2022). However, they overlook the length-wise bias issue induced by forcing all sequences in a batch to conform to a narrow length range. A more intrinsic relation between the

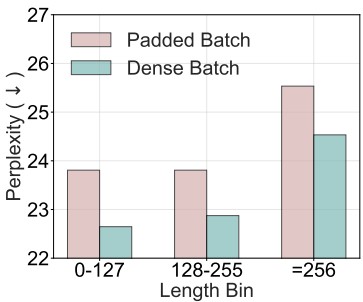

(a) Comparison of padded-/dense-batch training on perplexity

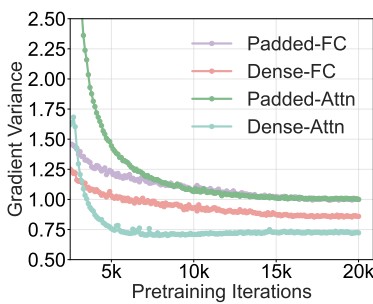

(b) Comparison of padded-/dense batch training on gradient variance

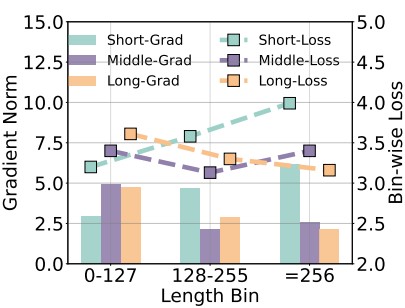

(c) The evalutation loss and gradient norm across different length bins.

Figure 3: **Empirical explorations** on length-aware data selection. (a) performance comparision on continual pretraining on different length bins after pretraining with variable-length padded batches and dense batches; (b) gradient variance comparison of models trained with variable-length padded batches and dense batches, with respect to the attention modules (referred to as Attn) and the last fully connected layer (referred to as FC); (c) relationship between evaluation loss and gradient norm across different length bins for models trained exclusively on a single length range. "Short", "Middle", and "Long" denote models trained on the $[0, 127]$, $[128, 255]$, and $[256]$ length bins, respectively. The curves ("∗-Loss") show the evaluation losses of these models on different length bins, while the bars ("∗-Grad") show the corresponding gradient norms. We leave the experimental details in Appendix A.3.2 for reference.

sequence length and token utilization efficiency, and an unbiased acceleration design with the underlying mechanism remain underexplored.

**Maximization of the TUR via dense batches.** The sequence length has a deterministic relationship with the TUR (defined in Eq. (1)) of a data batch under the padding preprocessing mechanism. In causal language modeling, the TUR is maximized when a data batch consists entirely of semantically continuous sequences of identical length, which we call *dense batches*. We find that *maximizing the TUR of data batches can benefit the model in learning foundational knowledge from training corpora*. Specifically, we pretrain a language model with regularly padded batches or dense batches for the same number of iterations and then continually pretrain it on the same data, sampled from a pre-defined length bin. As presented in Figure 3(a), dense batches enable the model to achieve lower perplexity across all length bins. To explain this merit, we examine the gradient variance dynamics during the training of the model with padded batches and dense batches. As presented in Figure 3(b), these dense batches yield significantly lower gradient variance, exhibiting the potential effect to speed up model convergence.

**Length-wise bias in single-length training.** To characterize the length bias issue of experiments in Figure 3(a), we partition training data into disjoint length bins, train separate models on each bin and evaluate them across all bins. In Figure 3(c), "Short", "Middle", and "Long" denote models trained on the $[0,127]$, $[128,255]$, and $[256]$ length bins, respectively, while the corresponding "∗-Loss" curves represent their evaluation losses across different length bins. As illustrated by the loss curves, the models exhibit a clear length-wise generalization gap: regardless of the chosen training bin, model performance degrades significantly on out-of-bin lengths compared to in-bin lengths. For instance, a model trained on the $[128-255]$ length bin achieves near-optimal in-bin performance, yet suffers severe performance degradation on both the shorter $[0-127]$ and the longer $[256]$ length bin. This aligns with recent findings in (Variš & Bojar, 2021; Anil et al., 2022). The details of this experiment can be found in Appendix A.3.2.

**Sampling probability adjustment via loss tracking.** To counteract the length-wise overfitting, we can actually build a training calibration through the sampling intervention. Specifically, prior works on importance sampling (Zhao & Zhang, 2015; Graves et al., 2017) demonstrates that the sampling distribution of training data should be roughly proportional to the gradient norms for faster convergence of stochastic optimization. Considering the computation cost of the gradient norm, we explore using the training loss as a surrogate metric. In Figure 3(c), we examine the relationship between gradient norm and loss across length bins. The bars labeled "∗-Grad" denote the gradient norms corresponding to the models trained on different length bins. We observe a consistent positive correlation between the gradient norm statistics ("∗-Grad") and

the corresponding evaluation losses ("*-Loss") across different length bins and models. Sequence-length bins associated with larger losses also tend to produce larger gradient norms, motivating us to utilize the training loss to adjust sampling probabilities for unbiased modeling on diverse length ranges. The experimental details are provided in Appendix A.3.2.

### 3.2 Dense-Balanced Sequence Scheduling

Based on the insights in Section 3.1, we aim to design a data scheduler that prioritizes uniform-length inputs to construct dense batches for maximization of token utilization in early training stages, and subsequently shifts to high-loss inputs to correct length-wise bias for balanced performance across varied length ranges. Specifically, we introduce the realization of our proposed online data selection method, Dense-Balanced Sequence Scheduling (DeBLAS), in detail, as illustrated in the right of Figure 2.

Before pretraining, we process the training corpus with the padding mechanism and then split it into $K$ bins based on sequence length. For simplicity, we set the length bins to $[0, \frac{L}{K-1}), [\frac{L}{K-1}, \frac{2L}{K-1}), \ldots, [\frac{(K-2)L}{K-1}, L), [L]$ with equally spacing, where $L$ denotes the model context length. We explore other strategies of length bin partition in Section 4.3. The overall data scheduling policy of DeBLAS can be formalized as a time-dependent sampling distribution $\pi_t(\boldsymbol{x})$ over the pretraining corpus $\mathcal{D}$ as follows:

$$\pi_t(x) \propto \begin{cases} \mathbb{I}(|\boldsymbol{x}| \geq L_{\mathrm{d}}), & 1 \leq t \leq T_d \\ P_k, & T_d < t \leq T \end{cases} \tag{2}$$

where $T_{\mathrm{d}}$ is the iteration for the stage transition, $T$ is the total number of training iterations, $L_{\mathrm{d}}$ is the uniform length for dense batching, and $P_k$ is the sampling probability of the bin $k$ that $\boldsymbol{x}$ belongs to.

**Stage I: Dense Batching.** At this stage ($t \leq T_d$), the model is trained exclusively on dense batches with a uniform sequence length $L_{\mathrm{d}}$. To be detailed, only sequences equal to or longer than $L_{\mathrm{d}}$ are sampled, truncated to $L_{\mathrm{d}}$ and then stacked together to construct dense batches. Therefore, every batch in the Dense Batching stage reaches the maximum TUR in the context of causal language modeling.

**Stage II: Balanced Batching.** At this stage ($T_d < t \leq T$), we randomly extract a small subset of training samples to craft a calibration set $D_{\mathrm{C}}$ which can approximately represent the length distribution of the whole training corpus, without noticeably reducing the training data size. The calibration set $D_C$ is completely disjoint from both the training batches, the validation set and all downstream evaluation data. During the training, the model is evaluated on $D_{\mathrm{C}}$ every $T_{\mathrm{C}}$ iterations, and the sampling probabilities for all the length bins are adjusted according to their respective evaluation losses and frequency proportions as follows:

$$P_k = \frac{r_k l_k}{\sum_{j=1}^{K} r_j l_j}, l_k = -\frac{1}{N_k} \sum_{j=1}^{N_k} \frac{1}{|\boldsymbol{x}^j|} \sum_{i=1}^{|\boldsymbol{x}^j|} \log P_\theta(x_i \mid x_{<i}), \tag{3}$$

where $N_k$ and $r_k$ indicate the sequence number and proportion of the $k^{\mathrm{th}}$ length bin and $l_k$ represent evaluation loss on the $k^{\mathrm{th}}$ length bin. By adaptively correcting the sampling probabilities of length bins according to their evaluation losses, DeBLAS achieves a balanced and debiased language modeling across diverse length ranges. Note that, this periodical calibration process adds minimal computational overhead compared to the whole training process, as empirically shown in Appendix A.3.3. The overall algorithm realization is summarized in Algorithm 1 of the appendix.

### 3.3 Theoretical Understanding

Here we present an idealized theoretical analysis of DeBLAS (full details and proofs in Appendix C) to support the intuition behind the proposed two-stage length-based scheduling strategy. First, as the Dense Batching stage prefers longer sequences $\pi_t(\boldsymbol{x}) \propto |\boldsymbol{x}|$ to involve more tokens into training, we can have the following lemma regarding the gradient variance reduction (Keskar et al., 2017).

**Assumption 1** (Token-Level Independence). *Let per-sample loss be $\ell(\boldsymbol{x}; \theta) = \sum_{i=1}^{|\boldsymbol{x}|} \log P_\theta(x_i \mid x_{<i})$, then assume that each token-level gradient $\nabla_\theta \log P_\theta(x_i \mid x_{<i})$ has bounded variance $\sigma_{\mathrm{tok}}^2$, and is independent across $i$ in a sequence sample $\boldsymbol{x} = (x_1, x_2, \ldots, x_{|\boldsymbol{x}|})$.*

**Lemma 1** (Variance Reduction by Long-Sequence Sampling). *The variance of the per-sequence gradient satisfies,* $\mathrm{Var}[\nabla\ell(\boldsymbol{x};\theta)] \leq \frac{\sigma_{tok}^2}{|\boldsymbol{x}|}$*, indicating longer sequences reduces the gradient variance.*

Subsequently, the Balanced Batching stage chooses samples with larger loss value $\pi_t(\boldsymbol{x}) \propto \ell(\boldsymbol{x};\theta_t)$, which realizes importance sampling (Zhao & Zhang, 2015) after focusing on dense batches in the previous stage and also brings variance reduction via amplifying the expected gradient norm.

**Assumption 2** (Gradient Magnitude Correlates with Loss). *Assume that the gradient norm and loss value are positively correlated for all $\boldsymbol{x}$ after the Dense Batching stage.*

**Lemma 2** (Gradient Norm Amplification by Loss-Based Sampling). *The loss-aware sampling increases the expected squared gradient norm:* $\mathbb{E}_{\boldsymbol{x}\sim\pi_t}\left[\|\nabla\ell(\boldsymbol{x};\theta)\|^2\right] \geq \mathbb{E}_{\boldsymbol{x}\sim\mathcal{D}}\left[\|\nabla\ell(\boldsymbol{x};\theta)\|^2\right]$.

**Theorem 1** (Convergence Acceleration via Two-Stage Curriculum). *Assume that Assumptions 1 and 2 hold, $\mathcal{L}(\theta)$ is L-smooth, and mini-batch gradients have bounded variance $\mathrm{Var}[\nabla\ell(\boldsymbol{x};\theta)] \leq \sigma^2$. Let stochastic gradient descent use the two-stage curriculum over $T = T_1 + T_2$ steps: $\boldsymbol{x} \sim \pi_t(\boldsymbol{x}) \propto |\boldsymbol{x}|$ if $t \leq T_1; \boldsymbol{x} \sim \pi_t(\boldsymbol{x}) \propto \ell(\boldsymbol{x};\theta_t)$ if $t > T_1$. Then the expected gradient norm satisfies*

$$\min_{0\leq t\leq T} \mathbb{E}[\|\nabla\mathcal{L}(\theta_t)\|^2] \leq \mathcal{O}\left(\frac{1}{\sqrt{T}}\right) - \eta \cdot \left(\Delta\sigma_{\text{length}}^2 + \Delta\sigma_{\text{loss}}^2\right),$$

*where $\Delta\sigma_{\text{length}}^2$ and $\Delta\sigma_{\text{loss}}^2$ represent the per-stage variance reduction compared to uniform sampling.*

**Remark 1.** For non-convex optimization using stochastic gradient descent, the average squared gradient norm decreases at a rate of $\mathcal{O}(1/\sqrt{T})$ (Carmon et al., 2018; Nesterov et al., 2018). This implies that to find a point where $\mathbb{E}[\|\nabla\mathcal{L}(\theta_t)\|^2] < \epsilon$, approximately $T = \mathcal{O}(1/\epsilon^2)$ iterations are required. Theorem 1 demonstrates a reduction in the convergence upper bound achieved by DeBLAS compared to standard training, which means that the model requires fewer iterations T to reach the same level of gradient norm. DeBLAS accelerates the convergence in LLM pretraining through its two-stage data scheduling: the *Dense Batching* stage achieves length maximization that enlarges $\Delta\sigma_{\text{length}}^2$, then the *Balanced Batching* stage further reduces the variance with positive $\Delta\sigma_{\text{loss}}^2$ via maximizing informative gradient signals. Intuitively, without the first stage, conducting high-loss selection includes noisy updates that prevent the convergence; without the second stage, only focusing on long-length samples can converge to near a suboptimal plateau. We empirically validate that both stages are indispensable in Section 4.3.

## 4 Experiment

### 4.1 Setup

**Datasets and evaluation.** In the main experiments, we employ C4 and SlimPajama as the pretraining corpora. Both of them are widely used to study LLM pretraining at a large scale. To evaluate the acceleration of pretraining, we measure the number of training iterations needed for a method to reach a given perplexity (PPL) on the validation set of C4 and Slimpajama. We additionally report end-to-end GPU hours as a practical runtime efficiency metric, which are measured based on the total pretraining wall-clock time on NVIDIA A100 GPUs. We also measure the performance of the models, which are pretrained to reach a given validation perplexity, on a comprehensive set of downstream benchmarks, including PIQA (Bisk et al., 2020), OpenBookQA (Mihaylov et al., 2018), Lambada-OpenAI (Paperno et al., 2016), Hellaswag (Zellers et al., 2019) and Arc-Easy (Clark et al., 2018) with accuracy as the evaluation metric. A detailed introduction of these datasets can be found in Appendix A.1.1.

**Baselines.** Since our method involves no additional model to assist with data selection, we compare our method with reference-model-free online data selection methods adapted for language model pretraining for a fair comparison. Besides random sampling (referred to as Random), we include Max Loss (Jiang et al., 2019), Max Grad Norm (referred to as Max Grad in Table 1) (Katharopoulos & Fleuret, 2018), Feature Matching (FM) (Bhatt et al., 2024), and GREATS (Wang et al., 2024), to select mini-batches with the same batch size as regular training from larger ones during pretraining. The selection ratios of these baseline methods are all set to 50%, which means the large sequence batch size is twice the base sequence batch

Table 1: Training Iterations (K, i.e., x1000) (↓) and end-to-end GPU hours (↓) required for different online selection methods in LLM pretraining to reach the given validation perplexity (PPL). The target PPL values are configured based on (Zhao et al., 2024a; Han et al., 2024). Values of relative speedup against Random ($T_{\text{Random}}/T_m$, ↑) in terms of iterations and GPU hours are also reported in brackets. Due to the high computational cost of LLaMA 1B model, we report ">R" for methods that fail to reach the target validation perplexity within the training iterations required by Random.

| Dataset | C4 | | | | SlimPajama | | | |
|---|---|---|---|---|---|---|---|---|
| Model size | 60M | 130M | 350M | 1B | 60M | 130M | 350M | 1B |
| Target PPL | 30.4 | 25.4 | 18.8 | 16.9 | 26.5 | 21.7 | 17.6 | 15.5 |
| | | | | Training Iterations | | | | |
| Random | 10 (-) | 20 (-) | 60 (-) | 100 (-) | 20 (-) | 40 (-) | 60 (-) | 100 (-) |
| Longest | 10 (1.00x) | 14 (1.43x) | 66 (0.91x) | 95 (1.05x) | 20 (1.00x) | 45 (0.89x) | 65 (0.92x) | >R |
| Max Loss | 14 (0.71x) | 28 (0.71x) | 84 (0.71x) | >R | 31 (0.65x) | 58 (0.69x) | 85 (0.70x) | >R |
| Max Grad | 12 (0.83x) | 17 (1.18x) | 83 (0.72x) | >R | 17 (1.18x) | 43 (0.93x) | 42 (1.43x) | >R |
| FM | 11 (0.91x) | 18 (1.11x) | 50 (1.20x) | >R | 17 (1.18x) | 50 (0.80x) | 48 (1.25x) | >R |
| GREATS | 12 (0.83x) | 17 (1.18x) | 66 (0.91x) | >R | 17 (1.18x) | 32 (1.25x) | 40 (1.50x) | 72.5 (1.34x) |
| **DeBLAS** | **8 (1.25x)** | **13 (1.54x)** | **40 (1.50x)** | **60 (1.67x)** | **14 (1.43x)** | **28 (1.43x)** | **39 (1.54x)** | **62.5 (1.60x)** |
| | | | | GPU Hours | | | | |
| Random | 1.56 (-) | 6.22 (-) | 65.12 (-) | 569.34 (-) | 3.19 (-) | 12.89 (-) | 65.98 (-) | 571.96 (-) |
| Longest | 1.56 (1.00×) | 4.35 (1.43×) | 71.63 (0.91×) | 540.87 (1.05×) | 3.19 (1.00×) | 14.50 (0.89×) | 71.48 (0.92×) | >R |
| Max Loss | 3.99 (0.39×) | 16.34 (0.38×) | 171.00 (0.38×) | >R | 8.97 (0.36×) | 35.06 (0.37×) | 175.13 (0.38×) | >R |
| Max Grad | 3.60 (0.43×) | 10.46 (0.59×) | 177.88 (0.37×) | >R | 4.98 (0.64×) | 26.85 (0.48×) | 91.28 (0.72×) | >R |
| FM | 3.38 (0.46×) | 11.34 (0.55×) | 109.89 (0.59×) | >R | 5.32 (0.60×) | 32.61 (0.40×) | 106.90 (0.62×) | >R |
| GREATS | 3.88 (0.40×) | 11.22 (0.55×) | 143.82 (0.45×) | >R | 5.58 (0.57×) | 21.89 (0.59×) | 93.48 (0.71×) | 907.87 (0.63×) |
| DeBLAS | **1.24 (1.26×)** | **4.02 (1.54×)** | **43.25 (1.51×)** | **341.04 (1.67×)** | **2.20 (1.45×)** | **9.02 (1.43×)** | **42.89 (1.54×)** | **357.47 (1.60×)** |

Table 2: Comparison between Random and DeBLAS in terms of the performance on a wide range of downstream tasks, which are evaluated using the model pretrained on C4 and SlimPajama with these two methods for the iterations required to reach the target validation perplexity in Table 1.

| Dataset | Model size | Method | Iterations (K) | Downstream tasks | | | | | Average |
|---|---|---|---|---|---|---|---|---|---|
| | | | | PIQA | OpenBookQA | Lambada | Hellaswag | ArcEasy | |
| C4 | 60M | Random | 10 | **60.3** | 23.6 | 13.4 | **28.0** | **35.4** | 32.1 |
| | | DeBLAS | 8 | 59.7 | **26.8** | **14.2** | 27.5 | 33.8 | **32.4** |
| | 130M | Random | 20 | **62.5** | **26.8** | 15.2 | 28.8 | 35.7 | 33.8 |
| | | DeBLAS | 13 | 61.5 | 26.4 | **16.2** | 28.8 | **36.5** | **33.9** |
| | 350M | Random | 60 | 65.3 | **29.4** | **23.7** | 33.1 | 38.3 | 38.0 |
| | | DeBLAS | 40 | **65.7** | 28.8 | 23.3 | **34.1** | **40.6** | **38.5** |
| | 1B | Random | 100 | 65.9 | 29.2 | 26.2 | 35.8 | 40.2 | 39.5 |
| | | DeBLAS | 60 | **66.6** | **30.0** | **26.5** | **36.9** | **41.8** | **40.4** |
| SlimPajama | 60M | Random | 20 | **58.3** | 23.8 | **14.0** | **27.5** | 35.3 | 31.2 |
| | | DeBLAS | 14 | 58.2 | **25.0** | 13.6 | 27.4 | **35.7** | **32.0** |
| | 130M | Random | 40 | **59.9** | **26.0** | 16.5 | **29.0** | 36.2 | 33.5 |
| | | DeBLAS | 28 | 59.5 | 25.6 | **17.9** | 28.7 | **36.4** | **33.6** |
| | 350M | Random | 60 | 62.4 | **28.2** | 20.5 | 31.5 | 39.3 | 36.4 |
| | | DeBLAS | 39 | **62.7** | 27.8 | **20.7** | **32.0** | **39.9** | **36.6** |
| | 1B | Random | 100 | **63.9** | 27.6 | 23.4 | 34.0 | 39.0 | 47.0 |
| | | DeBLAS | 62.5 | 63.5 | **28.4** | **23.8** | 34.0 | **41.9** | **47.9** |

size. In addition, we also conduct experiments on training the model with only sequences of length equal to the model max length $L$ (referred to as Longest in Table 1). We leave the introductions of the considered methods in Appendix A.1.2 and implementation details in Appendix A.2.

**Implementation details.** Regarding the hyperparameters of DeBLAS, we set the number of dense-batching iterations $T_d$ to 40% of the number of the iterations required for regular training to reach the target validation perplexity. We set the number of length bins $K$ to 3, the uniform length of the Dense Batching stage $L_d$ to half of the model context length, the calibration set size $N_C$ to 1000 and the calibration frequency $T_C$ to 1000 across all experiments in Table 1. We leave the details about reasons and guidelines for hyperparameter selection and additional sensitivity analyses of these hyperparameters in Appendix A.2.2.

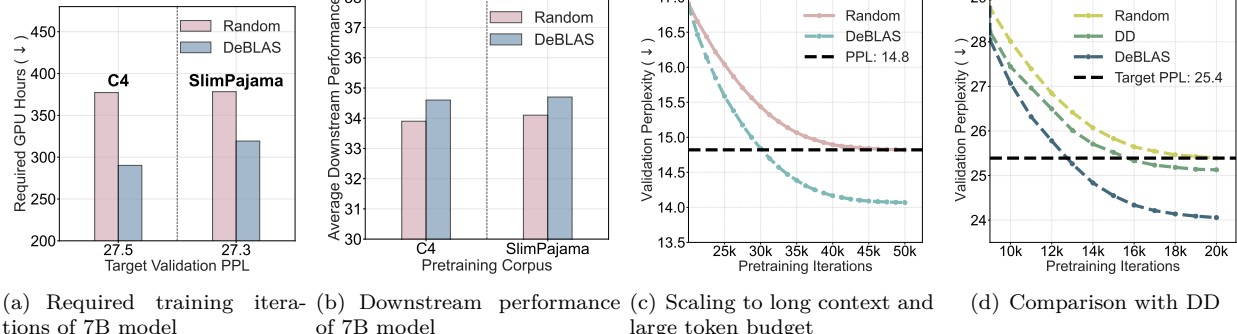

(a) Required training itera-
tions of 7B model

(b) Downstream performance
of 7B model

(c) Scaling to long context and
large token budget

(d) Comparison with DD

Figure 4: **Further evaluation**. (a) comparison on required GPU hours of 7B model to reach a given target validation perplexity; (b) comparison on downstream task performance on 7B model pretrained with iterations in Figure 4(a); (c) pretraining LLaMA 130M model on context length of 2048 and token budget of 100B, with corresponding end-to-end GPU hours reported in Appendix A.3.4; (d) comparison with Dataset Decomposition (DD) on pretraining efficiency.

## 4.2 Main Comparison on LLM Training Efficiency

**Evaluation on the validation set of the pretraining corpus.** As shown in Table 1, our method achieves consistent significant training iterations reduction across all model sizes and pretraining datasets. More importantly, the reduction in iterations achieved by DeBLAS translates almost directly into practical runtime acceleration. The close correspondence between iteration reduction and GPU-hour reduction is because DeBLAS introduces negligible additional per-iteration overhead. Notably, the performance gains scale with model size, demonstrating robustness to model size scaling. Methods utilizing simple model-related heuristics, such as training loss and gradient norm, exhibit inferior and unstable performance, occasionally even underperforming regular training. These results demonstrate that solely relying on model-related heuristics but ignoring intrinsic textual properties might not well capture the true informativeness of training data at current iterations. More critically, these methods incur substantially larger GPU-hour costs due to expensive candidate scoring and selection operations performed at each iteration. Additionally, although selecting only maximum-length sequences maximizes the token utilization rate, this strategy slows down model pretraining in most cases, which indicates the necessity of the Balanced Batching stage in our method.

**Evaluation on downstream tasks.** We evaluate the performance on a wide range of language modeling downstream benchmarks using the models pretrained on C4 with Random and our method for iterations required to reach the target perplexity in Table 1. As shown in Table 2, DeBLAS enables the model to reach the same level of downstream task performance on average in fewer iterations than Random across all model sizes, validating its efficiency and practicality.

**Scaling to 7B model.** Following the experiment configuration of (Han et al., 2024), we pretrain LLaMA 7B model with our method and only compare it with random sampling due to the resource constraints. As presented in Figure 4(a) and 4(b), DeBLAS achieves significant GPU hours reduction to reach the same level of validation perplexity and downstream task performance (averaged over 5 tasks in Table 2), demonstrating the effectiveness of our method under larger-scale model size.

**Scaling to long context and large token budget.** To evaluate DeBLAS on more practical settings of longer context length and larger token budget, we pretrain a LLaMA 130M model with SlimPajama using a context length of 2048 and a total of 100B training tokens. In order to scale our method to longer context and larger token budget, we design an adapted version of DeBLAS that maximizes both computation efficiency and data utilization. We present the detailed implementation in Algorithm 2 and a detailed explanation in Appendix A.3.4. As shown in Figure 4(c), DeBLAS can still outperform Random significantly.

**Comparison with Dataset Decomposition.** Dataset Decomposition (DD) (Pouransari et al., 2024) is a recent textual data preprocessing method that decomposes a given corpus containing documents of variable lengths into a collection of buckets, where every bucket $\mathcal{D}_i$ contains sequences of length $2^i$. At every training iteration, a bucket index $i$ is sampled based on a pre-defined curriculum to form a batch with $b/2^i$ sequences

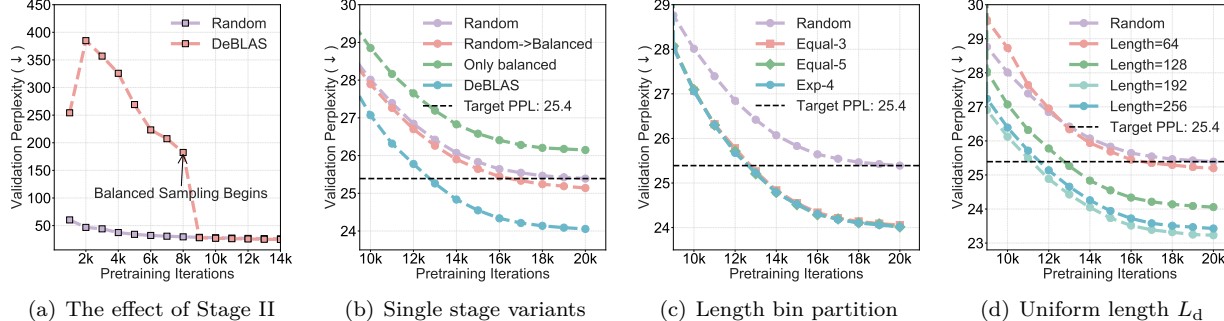

Figure 5: **Ablation study**. (a) LLaMA 130M pretrained on C4 with DeBLAS using uniform lengths $L_d = 128$; (b) pretraining efficiency comparison between DeBLAS and single-stage counterparts; (c) exploration of different length-bin partition strategies; (d) using different uniform lengths $L_d$ of the Dense Batching stage. We leave full experimental details to Appendix A.3.2.

from the bucket $D_i$, where $b$ is the fixed token batch size. Conceptually, DD confines the model to be exposed to a limited number of length choices (e.g. $2^i$), which may cause length-wise bias. In contrast, DeBLAS enables training across all possible sequence lengths at the Balanced Batching stages. We conduct an experiment to compare DD with its uniform curriculum and DeBLAS in terms of pretraining acceleration. As shown in Figure 4(d), DD shows no significant efficiency improvement over random selection, while our DeBLAS can achieve the target validation perplexity in much fewer iterations. A more in-depth exploration and experiment setups are presented in Appendix A.3.5.

## 4.3 Ablation Study and Further Discussion

In this section, we conduct further exploration and discussions to provide a thorough understanding of our method. We leave more experimental details and extra results in Appendix A.3.

**Importance of both stages.** Conceptually, without the Dense Batching stage, conducting high-loss selection includes gradient noises that slow down the convergence; without the Balanced Batching stage, training solely on uniform-length dense batches converges to a length-biased solution. As shown in Figure 5(a), the model performs poorly on the full validation set, which contains sequences of varying lengths, in the Dense Batching stage. The model exhibits a rapid decline in perplexity after the Balanced Batching stage begins and eventually surpasses Random. Furthermore, in Figure 5(b), we illustrate the performance comparison of our method with two single-stage counterparts. The first variant employs only the Balanced Batching stage throughout the pretraining (denoted as "Only balanced"). The second variant performs standard random sampling during the first 40% of training iterations and then switches to the Balanced Batching stage (denoted as "Random->Balanced"). The results show that both variants significantly underperform DeBLAS, verifying the necessity of the Dense Batching stage for effective acceleration of LLM pretraining.

**Sampling methods of the second stage.** To further validate the importance of dynamic sampling probabilities for length bins in the Balanced Sampling stage, we compare DeBLAS against a variant, DeBLAS-Uniform, where the Balanced Batching stage samples from all length bins with equal probabilities ($P_k = 1/K$ for all $k \in [1, 2, \cdots .K]$). As shown in Table 3, although DeBLAS-Uniform improves over Random, it converges more slowly than

Table 3: Comparison between DeBLAS-Uniform and DeBLAS with C4 dataset. The number of iterations (in thousands) required to reach the target PPL and relative speedup are reported.

| Model size | Target PPL | Method | | |
|---|---|---|---|---|
| | | Random | DeBLAS-Uniform | DeBLAS |
| 60M | 30.4 | 10 (-) | 9 (1.11x) | 8 (1.25x) |
| 130M | 25.4 | 20 (-) | 17 (1.18x) | 13 (1.54x) |

standard DeBLAS. Sampling with equal probabilities or fixed curricula in the Balanced Batching stage might waste computational budget on data that the model has already been heavily exposed to in the Dense Batching stage. Dynamic sampling allows the model to allocate the majority of its budget to the underrepresented lengths, achieving the target perplexity with fewer pretraining iterations.

**Influence of length bin partition.** In this part, we explore other alternatives for length bin partition with LLaMA 130M models pretrained on C4, including denser equally spaced partitions and exponentially uneven partitions. To be specific, we consider equally spaced 3 bins, equally spaced 5 bins and exponentially spaced 4 bins, which are referred to as Equal-3, Equal-5 and Exp-4, respectively, in Figure 5(c). As presented by the results, our method exhibits excellent acceleration performance across all the considered partitioning schemes, highlighting its robustness to different partition strategies.

**Generality regarding the uniform length of dense batches.** We investigate the influence of $L_\mathrm{d}$, the uniform length of in the Dense Batching stage, in Figure 5(d) with LLaMA 130M models pretrained on C4. The token batch size remains fixed with varying $L_\mathrm{d}$. The results reveal that assigning $L_\mathrm{d}$ to a value greater than the half of the model max length (i.e., 256), yields strong performance, while excessively reducing this value (e.g. to 64) results in significant performance degradation, likely due to the model's inability to develop coherent representations of longer linguistic structures from such short sequences.

**Comparison with reference-model-based methods.** We conduct experiments on two representative reference-model-based online data selection methods, RHO-LOSS (Mindermann et al., 2022) and Bayesian Data Selection (BDS) (Deng et al., 2023). We use the LLaMA 60M model pretrained with the Random baseline, which is presented in Table 1, as the reference model and set the selection ratio to 50%. We pretrain LLaMA 130M models on C4 to reach the target validation perplexity of 25.4, same as in Table 1. As presented in Table 4, our DeBLAS requires substantially fewer training iterations than these two methods.

Table 4: Comparison between RHO-LOSS, BDS and DeBLAS, evaluated on an NVIDIA A100 GPU. The number of iterations (in thousands) required to reach the target PPL is reported.

| Method | Required iterations | Time per iter (in seconds) | Time in total (in hours) |
|---|---|---|---|
| Random | 20 | 1.12 | 6.22 |
| RHO-LOSS | 18 | 2.78 | 13.90 |
| BDS | 18 | 2.88 | 14.40 |
| DeBLAS | 13 | 1.11 | 4.02 |

Furthermore, RHO-LOSS and BDS incur considerable time overhead at every iteration because both the reference model and the current model must perform forward passes on all candidate sequences. An introduction to RHO-LOSS and BDS and experimental details can be found in Appendix A.3.6

**Memory efficiency.** Conceptually, DeBLAS does not increase the overall token budget per iteration. Instead, it improves training efficiency by scheduling sequence length and enhancing token utilization. Therefore, its GPU memory consumption should be comparable against the Random baseline. We evaluate the memory efficiency of the two stages of DeBLAS by profiling GPU utilization and peak GPU memory consumption on a single NVIDIA A100 GPU with LLaMA 130M model and C4 dataset, following the same configuration used in Table 1. As shown in Table 5, both stages achieve comparable GPU utilization and peak GPU memory usage against Random, indicating that the online calibration and adaptive sampling of the Balanced Batching stage introduce only negligible runtime overhead and memory consumption. These results suggest that the acceleration gains of DeBLAS are achieved without materially increasing GPU memory consumption or sacrificing hardware efficiency. The details of this profiling experiment are provided in Appendix A.3.7.

Table 5: Memory efficiency analysis of DeBLAS compared against Random.

| Method | GPU utilization (%) | Peak GPU Memory (GB) |
|---|---|---|
| Random | 90.26 | 46.21 |
| DeBLAS-Dense | 90.32 | 46.10 |
| DeBLAS-Balanced | 89.35 | 46.88 |

**Stability of learned length-bin sampling probabilities.** To investigate whether the learned sampling behavior of the Balanced Batching stage remains stable across different random seeds, datasets, and model scales, we evaluate DeBLAS with three random seeds on C4 and SlimPajama using LLaMA models ranging from 60M to 350M parameters, with other hyperparameters the same as in Table 1. For each configuration, we record the sampling probability of every length bin at each calibration step throughout the Balanced Batching stage, compute the standard deviation of the sampling probabilities across random seeds, and then average the result over all length bins and calibration steps. The results of Table 6 show consistently low variability across all datasets and model sizes, indicating that the calibration mechanism learns stable length-wise balancing behaviors. The experimental details are provided in Appendix A.3.8.

Table 6: Average standard deviation of learned length-bin sampling probabilities across random seeds, computed over all length bins and calibration steps.

| Dataset | Model size | | |
|---|---|---|---|
| | 60M | 130M | 350M |
| C4 | 0.013 | 0.011 | 0.012 |
| SlimPajama | 0.016 | 0.012 | 0.014 |

**Discussion on other data attributes.** There are other common data attributes such as semantic difficulty and domain distribution. However, semantic difficulty lacks a reliable and universally applicable metric, and detailed domain annotations are often unavailable or prohibitively expensive to obtain at scale, which makes it difficult to incorporate such attributes into a controllable data scheduling framework. Nonetheless, DeBLAS already implicitly accounts for semantic difficulty. During the Balanced Batching stage, length bins are reweighted based on their evaluation losses, which serve as a commonly used proxy for semantic difficulty in practice. Furthermore, to validate the generalization capabilities of DeBLAS, we evaluate models pretrained with Random and DeBLAS on the validation data of different domains in SlimPajama. As shown in Table 15, DeBLAS achieves comparable or lower perplexity than the Random baseline across different domains, indicating that our method generalizes well in the presence of diverse semantic distributions. A more detailed discussion can be found in Appendix A.3.9.

## 5   Conclusion

In this paper, we propose a two-stage online data selection method, i.e., Dense-Balanced Sequence Scheduling (DeBLAS), for accelerating LLM pretraining that strategically transitions from dense-batched sequences to high-loss training instances. Delving into the core utilization on the sequence tokens in language modeling, our method is motivated by a dual goal of enhancing early-stage representation richness and later-stage length-wise generalization. The first stage prioritizes token-rich inputs to encourage a broad coverage of vocabulary and structural patterns, while the second stage shifts to loss-aware sampling to refine the training trajectory. Our theoretical analysis suggests that both stages can reduce gradient variance to reach a lower convergence bound. Empirical verifications are conducted for different model sizes across various datasets. We hope this principled and practical enhancement for accelerating LLM pretraining can bring new insights for future work on adaptive schedule learning or fine-grained data difficulty estimation.

### Broader Impact Statement

While our method focuses on length-based scheduling rather than explicit semantic filtering or content modification, length-based scheduling may still implicitly alter the effective distribution of training data if sequence length correlates with factors such as domain, source, writing style, or semantic characteristics. In this work, we partially examine this issue through domain-level evaluations on SlimPajama, where DeBLAS demonstrates comparable perplexity across different domains against random sampling. Nevertheless, a more comprehensive investigation of the interaction between sequence length, domain distribution, and potential representation bias remains an important direction for future work.

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

## A    Complete Experiment Details

### A.1    Details about Considered Baselines and Datasets

In this section, we provide the details about the training and evaluation datasets as well as the baselines for reference-model-free online data selection that are considered in our experiment section.

### A.1.1    Datasets

**Pretraining corpora.** C4[1] dataset is a colossal, cleaned version of Common Crawl's web crawl corpus (Raffel et al., 2020), which was initially used to train the T5 text-to-text Transformer models. It consists of approximately 750GB of clean English text scraped from the web. SlimPajama[2] is a high-quality text corpus curated for pre-training large language models. It originates from RedPajama (Weber et al., 2024), an open-source research project that is designed to replicate the pretraining data of Llama (Touvron et al., 2023) and contains over 1.2 trillion tokens. Through rigorous filtering to eliminate redundant and low-quality text, SlimPajama retains only 50% of the original tokens from RedPajama.

**Downstream benchmarks.** We evaluate pretrained models in a zero-shot manner on a comprehensive set of commonsense reasoning benchmarks from the widely recognized evaluation framework, Harness (Gao et al., 2024)[3], including PIQA (Bisk et al., 2020), OpenBookQA (Mihaylov et al., 2018), Lambada-OpenAI (Paperno et al., 2016), Hellaswag (Zellers et al., 2019) and Arc-Easy (Clark et al., 2018). **Physical Interaction**

---

[1]https://huggingface.co/datasets/allenai/c4
[2]https://huggingface.co/datasets/MBZUAI-LLM/SlimPajama-627B-DC
[3]https://github.com/EleutherAI/lm-evaluation-harness

**Question Answering (PIQA)** (Bisk et al., 2020) is a physical commonsense reasoning benchmark designed to investigate the physical knowledge of language models, which consists of 1,838 2-choice questions. **Open-BookQA** (Mihaylov et al., 2018) is a question-answering dataset aimed at measuring basic physical and scientific intuition about common objects and entities. It consists of 5,957 four-choice elementary-level science questions (4,957 train, 500 dev, 500 test), which probe the understanding of a small "book" of 1,326 core science facts and the application of these facts to novel situations. **Lambada-OpenAI** (Paperno et al., 2016) is a word-prediction task to test the model capability for language understanding. It consists of 5,153 passages extracted from books and the model is expected to read the first $N - 1$ words of each passage and predict the final token. To succeed on Lambada-OpenAI, models cannot simply rely on local context, but must be able to keep track of information in the broader discourse. **Hellaswag** (Zellers et al., 2019) is a challenging benchmark designed to assess language models' commonsense reasoning ability and consists of 10,042 four-choice questions in which the model is prompted with a scenario and chooses the most likely conclusion to the scenario from four possible options. **AI2 Reasoning Challenge (ARC)** is a question-answering benchmark (Clark et al., 2018) that evaluates commonsense knowledge and reasoning capabilities and we use the easy set which consists of 5,197 easy four-choice science questions drawn from grade 3-9 science exams.

### A.1.2 Baselines

**Hard-mining-based online batch selection.** (Jiang et al., 2019; Katharopoulos & Fleuret, 2018; Loshchilov & Hutter, 2015) prioritize hard samples from a large candidate batch based on heuristic criteria. **Max Loss** selects top-K data with the highest training losses and **Max Grad Norm** selects top-K data with the highest gradient norm. In our experiment, we leverage the gradient of the loss with respect to the final layer of the model to implement Max Grad Norm, following the practice of (Katharopoulos & Fleuret, 2018; Hong et al., 2024).

**Feature Matching.** (Bhatt et al., 2024) employs the framework of submodular Facility Location (FL) function optimization to select the most informative samples to label in order to mitigate the annotation cost of supervised fine-tuning of LLMs. The facility location problem is defined as follows,

$$S = \underset{S \subset X, |S|=k}{\arg\max} \sum_{x_i \in X} \max_{x_j \in S} w_{ij} \tag{4}$$

where $X$ denotes the unlabeled data pool, $k$ denotes the number of selected data and $w_{ij}$ denotes the similarity score between the features of data points $x_i$ and $x_j$. This approach can be easily adapted for online data selection of LLM pretraining. Following (Bhatt et al., 2024), we use the hidden state output by the last layer of the transformer-based language model as the feature and use $w_{ij} = \exp(-\|f(x_i) - f(x_j)\|)$.

**GREATS.** (Wang et al., 2024) formulates the online batch selection problem as a set utility function optimization task as follows,

$$\widehat{\mathcal{B}}_t^{(k)} = \underset{S \subseteq \mathcal{B}_t, |S|=k}{\arg\max} U^{(t)}(S) \tag{5}$$

$$U^{(t)}(S; Z^{(\text{val})}) := \ell(w_t, Z^{(\text{val})}) - \ell(\widetilde{w}_{t+1}(S), Z^{(\text{val})}) \tag{6}$$

where the utility function $U^{(t)}(S)$ quantifies how much a training data subset $S$, chosen from a large candidate batch $\mathcal{B}_t$, reduces the model's loss on a small given target-domain validation set $Z^{(\text{val})}$, directly linking batch selection to validation performance. To eliminate the need for expensive model updates and validation loss evaluations for each candidate subset, GREATS solves the set utility function optimization problem via the greedy algorithm and leverages Taylor expansions to approximate the impact of a training example on the model's validation loss using gradient inner-products between the training examples and the validation data. It also utilizes the "ghost inner-product" technique for efficient computation of pairwise gradient inner-products without the need to instantiate any model-sized vectors. Under the pretraining setting, the validation data are held out from the pretraining corpus.

Table 7: Hyperparameters of LLaMA model across different sizes.

| Parameter name | 60M | 130M | 350M | 1B | 7B |
|---|---|---|---|---|---|
| Parameter count | 58073600 | 134105856 | 367969280 | 1339082752 | 6738415616 |
| Hidden size | 512 | 768 | 1024 | 2048 | 4096 |
| Intermediate hidden size | 1376 | 2048 | 2736 | 5461 | 11008 |
| Attention head number | 8 | 12 | 16 | 32 | 32 |
| Layer number | 8 | 12 | 24 | 24 | 32 |
| Vocabulary size | 32000 | 32000 | 32000 | 32000 | 32000 |
| Minimum learning rate | 2.5e-4 | 2.5e-4 | 1e-4 | 5e-5 | 5e-5 |
| Maximum learning rate | 2.5e-3 | 2.5e-3 | 1e-3 | 5e-4 | 5e-4 |
| Warmup iteration number | 1000 | 2000 | 6000 | 10000 | 1500 |
| Gradient clipping | 0.0 | 0.0 | 0.0 | 0.0 | 1.0 |

Table 8: General training hyperparameters for C4 and Slimpajama experiments.

| Dataset | Parameter name | 60M | 130M | 350M | 1B | 7B |
|---|---|---|---|---|---|---|
| | Minimum learning rate | 2.5e-4 | 2.5e-4 | 1e-4 | 5e-5 | 5e-5 |
| | Maximum learning rate | 2.5e-3 | 2.5e-3 | 1e-3 | 5e-4 | 5e-4 |
| C4 | Iterations for regular training | 10000 | 20000 | 60000 | 100000 | 13000 |
| | Warmup iteration number | 1000 | 2000 | 6000 | 10000 | 1350 |
| | Gradient clipping | 0.0 | 0.0 | 0.0 | 0.0 | 1.0 |
| | Minimum learning rate | 2.5e-4 | 1e-4 | 1e-4 | 5e-5 | 5.7e-5 |
| | Maximum learning rate | 2.5e-3 | 1e-3 | 1e-3 | 5e-4 | 5e-4 |
| Slimpajama | Iterations for regular training | 20000 | 40000 | 60000 | 100000 | 13000 |
| | Warmup iteration number | 2000 | 4000 | 6000 | 10000 | 1500 |
| | Gradient clipping | 0.0 | 0.0 | 0.0 | 0.0 | 1.0 |

## A.2 Details about model architectures and hyperparameters

### A.2.1 Details about model architectures and general pretraining hyperparameters

In this section, we introduce the details of the model architecture and hyperparameters used for the main experiments. Following many previous works (Lialin et al., 2023; Zhao et al., 2024a; Han et al., 2024), we adopts a LLaMA-based model architecture (Touvron et al., 2023; Grattafiori et al., 2024) with pre-normalization, RMSNorm (Zhang & Sennrich, 2019) and SwiGLU activations (Shazeer, 2020). We consider varying model sizes ranging from 60M up to 7B parameters. For each model size, we use the same set of hyperparameters across all considered methods. Specific hyperparameters are shown in Table 7 and Table 8. We use the Adam optimizer (Kingma & Ba, 2014) with $\beta_1 = 0.9$, $\beta_2 = 0.999$ and no weight decay. We use a max sequence length of 256 for C4 and 512 for SlimPajama for all models, with a fixed token batch size of 131K tokens. For all experiments, we adopt learning rate warmup in the beginning of training and use cosine annealing for the learning rate schedule, decaying to 10% of the initial learning rate. We train the model with the BFloat16 format to reduce memory usage. The numbers of total training iterations $T$ for regular training (Random) on C4 align with those used in (Zhao et al., 2024a; Han et al., 2024). The numbers of total training iterations $T$ for regular training (Random) on Slimpajama with LLaMA 60M and 130M model are twice that on C4 since we observe these two models don't converge within the iterations for C4. To best simulate the practical pre-training scenario, we train without data repetition over a large amount of data. We evaluate the model every 1000 iterations for LLaMA 60M, 130M and 350M and every 2500 iterations for LLaMA 1B. All experiments are run on NVIDIA A100 (80 GB) GPUs with Python 3.10 and PyTorch 2.5.0.

---

**Algorithm 1** Dense-Balanced Sequence Scheduling (DeBLAS)

---

**Require:** learning rate $\eta$, base batch size $B$, total pretraining iterations $T$, Dense Batching stage iterations $T_d$, model max length $L$, number of length bins $K$, uniform length of dense batch $L_d$, calibration set size $N_C$, calibration frequency $T_C$

**Ensure:** pretrained model $\boldsymbol{\theta}_T$

1: Define length bins: $[0, \frac{L}{K-1}), [\frac{L}{K-1}, \frac{2L}{K-1}), \ldots, [\frac{(K-2)L}{K-1}, L), [L]$
2: Randomly sample $N_C$ sequences from the training set to construct a calibration set $D_{\mathrm{C}}$ and compute the length ratio vector $\boldsymbol{r}_C = [r_1, r_2, \cdots, r_K]$
3: $B_d \leftarrow \frac{L}{L_d} B$
4: **for** $t = 1$ to $T_d$ **do**
5:      Sample sequences longer than $L_d$ and truncate them to length $L_d$
6:      Construct a dense batch $\{\boldsymbol{x}^{(i)}\}_{i=1}^{B_d}$
7:      $\boldsymbol{\theta_t} \leftarrow \boldsymbol{\theta_{t-1}} - \eta \nabla_{\boldsymbol{\theta_{t-1}}} \left\{ \ell(\{\boldsymbol{x}^{(i)}\}_{i=1}^{B_d}, \boldsymbol{\theta}_{t-1}) \right\}$
8: **end for**
9: **for** $t = T_d + 1$ to $T$ **do**
10:      **if** $(t - T_d) \bmod T_C = 0$ **then**
11:         Evaluate $\theta_{t-1}$ on $D_{\mathrm{C}}$, yielding losses on $K$ bins $\boldsymbol{l}_C = [l_1, l_2, \cdots, l_K]$
12:         $P_k \leftarrow \frac{r_k l_k}{\sum_{j=1}^{K} r_j l_j}$
13:      **end if**
14:      Sample a batch $\{\boldsymbol{x}^{(i)}\}_{i=1}^{B}$ according to $[P_1, P_2, \cdots, P_K]$
15:      $\boldsymbol{\theta_t} \leftarrow \boldsymbol{\theta_{t-1}} - \eta \nabla_{\boldsymbol{\theta_{t-1}}} \left\{ \ell(\{\boldsymbol{x}^{(i)}\}_{i=1}^{B_d}, \boldsymbol{\theta}_{t-1}) \right\}$
16: **end for**

---

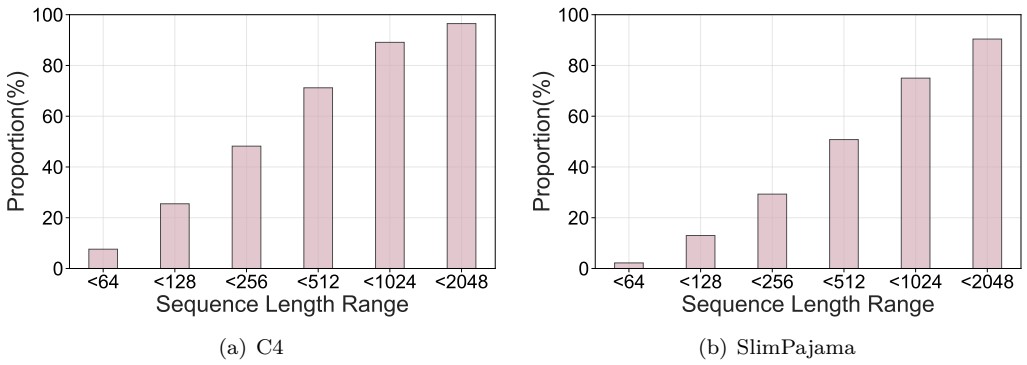

          (a) C4                              (b) SlimPajama

Figure 6: (a) The proportions of different sequence length ranges of C4; (b) The proportions of different sequence length ranges of SlimPajama. They are both counted in sequence number.

### A.2.2   Details about hyperparameters of DeBLAS

We configure the number of dense-batching iterations $T_d$ to 40% of the number of the iterations required for regular training to reach the target validation perplexity, the number of length bins $K$ to 3, the uniform length of the Dense Batching stage $L_d$ to half of the model max sequence length (i.e., 128 for C4 and 256 for Slimpajama), the calibration set size $N_C$ to 1000 and the calibration frequency $T_C$ to 1000 across all experiments in Table 1.

**Dense Batching stage iterations $T_d$.** The transition timing between the Dense Batching and Balanced Batching stage is a crucial hyperparameter. An excessively short Dense Batching stage fails to fully exploit the acceleration potential of the maximized token utilization of the dense batch. Conversely, an overly long Dense Batching stage introduces a more severe length bias, which then requires a longer Balanced Batching stage to counteract and ultimately results in more training iterations to reach a given validation perplexity.

Intuitively, it's a reasonable choice to make the transition when the training loss of the Dense Batching stage begins to plateau. Specifically, we record the training loss at intervals of 5% of the total training iterations and define the start of a plateau as a loss difference of less than 0.1 between the current and the last recorded iteration. In the main experiments of this work, we set the number of Dense Batching iterations $T_d$ to 40% of the number of iterations required for regular training to reach the target validation perplexity.

**Number of length bins (K).** Figure 5(c) compares multiple length bin partition strategies (Equal-3, Equal-5, Exp-4). The results show that DeBLAS is robust across different binning schemes, suggesting that the method is not sensitive to the exact choice of K. Empirically, we find that using 3-5 bins provides a good trade-off between calibration granularity and implementation simplicity.

**Uniform length of the Dense Batching stage $L_d$.** We consider $L_d$ from two primary perspectives. Firstly, if $L_d$ is too short, such as set to 1/4 of the model context length, the model is forced to learn from short contexts, which severely limits the attention mechanism's ability to capture long-range dependencies in the initial training stage. Secondly, the choice of $L_d$ should align with the natural length distribution of the pretraining corpus. If $L_d$ is significantly shorter than the dominant length range of the pretraining corpus, the majority of documents are aggressively truncated, breaking their semantic continuity. On the contrary, if the $L_d$ is significantly larger than the dominant length range, there won't be enough sequences to form the dense batches required by the Dense Batching stage. As shown in Figure 6, a substantial portion of documents in C4 fall within the $[128, 256]$ token range, which ensures that the dense batches encapsulate complete or near-complete linguistic patterns. In general, $L_d$ should be chosen large enough to capture meaningful long-context dependencies and aligns the natural length distribution of the pretraining corpus. If the model context length $L$ aligns with the natural length distribution of the pretraining corpus, a choice of $L_d$ from a range of $[L/2, L]$ would be recommended.

**Calibration set size $N_C$.** Conceptually, the calibration set $D_C$ only needs to approximate the global length distribution of the pretraining corpus. We conduct sensitivity experiments with different $N_C$ while keeping all other hyperparameters fixed, following the same experimental setup as Figure 5. As shown in Table 9, DeBLAS is not particularly sensitive to the exact calibration set size once a moderate number of samples is used. A relatively small subset (e.g., around 1K samples) already provides good acceleration performance.

Table 9: Sensitivity analysis of $N_C$ with LLaMA 130M pretrained on C4. The number of required iterations (K, i.e., x1000) to reach the target PPL of 25.4 are reported.

| $N_C$ | 200 | 1000 | 2000 | 5000 |
|---|---|---|---|---|
| Iterations | 16 | 13 | 13 | 13 |

**Calibration frequency ($T_C$).** Conceptually, very frequent recalibration increases evaluation overhead without noticeable performance gains, whereas overly infrequent updates slow the adaptation to length-wise bias. Empirically, we perform additional ablations on the calibration frequency $T_C$ in Table 10, following the same experimental setup as Figure 5. The results show that DeBLAS is similarly robust to the calibration frequency. Updating every 1K iterations already achieves a favorable balance between convergence performance and computational efficiency.

Table 10: Sensitivity analysis of $T_C$ with LLaMA 130M pretrained on C4. The number of required iterations (K, i.e., x1000) to reach the target PPL of 25.4 are reported.

| $T_C$ | 500 | 1000 | 2000 | 4000 |
|---|---|---|---|---|
| Iterations | 13 | 13 | 14 | 16 |

### A.3 Additional Experimental Results and Further Discussion

In this section, we provide more experiment results from various perspectives to characterize our proposed method. Firstly, we discuss the difference between the two main data preprocessing methods in language modeling, packing and padding. Secondly, we clarify the distinctions between DeBLAS and Dataset Decomposition (Pouransari et al., 2024), a related work on sequence length curriculum. Thirdly, we introduce the additional experimental setups for the empirical verification in the previous figures and our learning framework. Lastly, we present the evaluation results of our method on other large-scale text datasets.

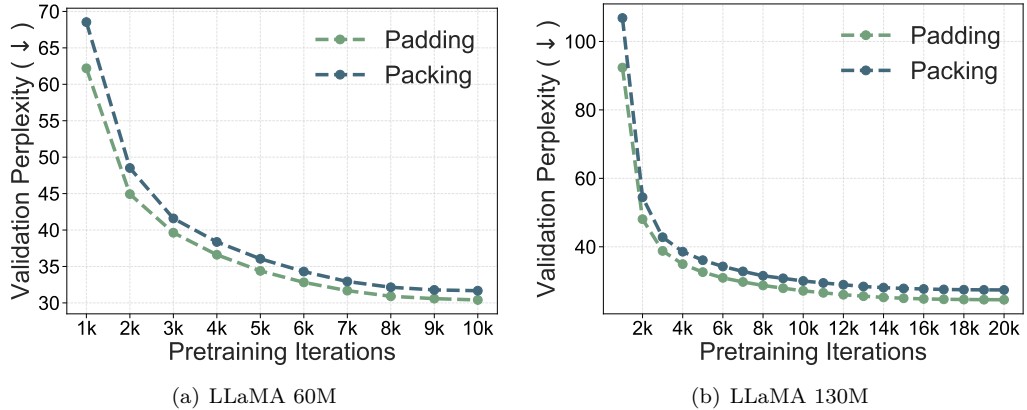

(a) LLaMA 60M (b) LLaMA 130M

Figure 7: Validation perplexity over iterations of LLaMA 60M and 130M models pretrained with packing and padding on C4.

### A.3.1 Discussion on packing

Although packing reduces the wasted compute induced by the padding tokens, it harms data integrity when whole documents are fragmented into independent segments and segments of different documents are packed into the same sequence. This naturally results in loss of information, reduces the effective context length, and thus makes models more prone to hallucination (Ding et al., 2024; Zhao et al., 2024b). To evaluate the computational trade-offs between sequence packing and padding, we conduct an experimental comparison on the convergence of language model pretraining. The packing implementation follows the common practice adopted in modern LLM pretraining, where document boundaries are preserved through attention masks to avoid cross-document contamination. We pretrain a LLaMA 60M and a 130M model with C4, using the hyperparameter configuration in Appendix A.2, on batches preprocessed via packing and padding (referred to as packed batch and padded batch), respectively. As shown in Figure 7, packing causes the model to train for a greater number of iterations to achieve the same validation perplexity as padding under a fixed token batch size, suggesting that the reduction of padding tokens in packed batches does not necessarily translate into faster optimization convergence. In contrast, DeBLAS accelerates pretraining from the perspective of both token utilization and model generalization. The Dense Batching stage improves TUR by reducing unnecessary padding within batches, while the Balanced Batching stage further improves optimization efficiency and mitigates length-wise bias through adaptive loss-aware scheduling across all sequence lengths.

Additionally, we conduct experiments on both C4 and SlimPajama using LLaMA 60M and 130M models to directly compare packing and DeBLAS. As shown in Table 11, DeBLAS consistently achieve better training efficiency than packing. We conjecture that the document fragmentation introduced by packing is an important factor for this performance gap. Although attention masks successfully prevent cross-document contamination, documents are frequently split across multiple packed sequences. Con-

Table 11: Training Iterations (x1000) (↓) required for Packing and DeBLAS to reach the given validation perplexity (PPL). Values of relative speedup against Packing ($T_{\text{Packing}}/T_m$, ↑) are reported in brackets.

| Dataset | C4 | | SlimPajama | |
|---|---|---|---|---|
| Model size | 60M | 130M | 60M | 130M |
| Target PPL | 30.4 | 25.4 | 26.5 | 21.7 |
| Packing | 11 (-) | 22 (-) | 21 (-) | 41 (-) |
| **DeBLAS** | **8 (1.38x)** | **13 (1.69x)** | **14 (1.50x)** | **28 (1.46x)** |

sequently, many tokens lose access to their original long-range context, which interrupts semantic continuity and weakens long-context dependency modeling. This observation is also consistent with our token utilization rate (TUR) formulation in Eq. (1). After document splitting, the number of semantically meaningful preceding tokens that each token can attend to decreases substantially, leading to lower TUR. Therefore, while packing eliminates padding tokens, it may still reduce optimization efficiency compared to DeBLAS.

### A.3.2 Additional experimental setups

**Figure 3.** In Figure 3(a), we conduct a preliminary experiment to investigate the impact of dense batches on pretraining efficiency with a LLaMA 130M model on C4 dataset. We set the model max length $L = 256$, the token batch size to $N_B = 256 \times 512 = 131072$ and split the length range of C4 into three bins, i.e, $[0, 127], [128, 255], [256]$. All sequences longer than 256 are truncated to 256 and assigned to the $[256]$ bin. We first pretrain two LLaMA 130M models with regularly padded batches and dense batches, respectively, for 10000 iterations. The uniform length of the dense batches is configured to 128. We then continually pretrain them with the same set of data sampled from a single length bin for another 10000 iterations and finally evaluate them on the validation data from the continually pretrained length bin. In Figure 3(b), we pretrain two LLaMA 130M models with regularly padded batches and dense batches, respectively, for 20000 iterations and calculate the gradient variance over sliding windows of the last 100 iterations, sampled at intervals of every 100 iterations. Specifically, we consider the gradients with respect to the last fully connected layer and the attention modules of all the layers. For the attention modules, we concatenate the gradient matrices of all the attention modules together to calculate the the empirical mean and variance. In Figure 3(c), we split the length range of C4 into three bins, i.e. $[0, 127], [128, 255], [256]$ and pretrain three LLaMA 130M models for 20000 iterations with data sampled from the three length bins, respectively, and then evaluate on the original validation set of C4. In the figure legend, "Short", "Middle", and "Long" denote models trained on the $[0, 127], [128, 255]$, and $[256]$ length bins, respectively. The curves labeled "Short-Loss", "Middle-Loss", and "Long-Loss" represent the evaluation losses of these three models on different length bins. We continually calculate the three models' gradients of loss, respectively, on three sets of regularly padded batches, each of which is sampled from one of the three length bins and has no overlapping sequences with the pretrained data, with respect to the final fully connected layer of the model and plot the average results in Figure 3(c). The gradient norm is computed as the gradient norms of all model parameters averaged over a randomly-sampled subset from a length bin of the C4 with a size of 1 million tokens. The bars labeled "Short-Grad", "Middle-Grad", and "Long-Grad" represent the corresponding gradient norms for these three models calculated on different length bins.

**Figure 5.** In Figure 5(c), we explore two other alternatives of length-bin partition, namely, Equal-5 and Exp-4. The model max length is set to $L=256$. Equal-5 denotes partitioning the length ranges into 5 equally spaced bins except the last bins, which are $[0, 63], [64, 127], [128, 195], [196, 255], [256]$. Exp-4 denotes partitioning the length range into 4 exponentially uneven bins, which are $[0, 63], [64, 127], [128, 255], [256]$. The average sequence length of C4 when using Random is 196.4, based on our measurement.

**Training details.** Following the configuration in Zhao et al. (2024a); Han et al. (2024), we conduct all the pretraining experiments in Figure 3 and Figure 5 with ADAM (Kingma & Ba, 2014) optimizer, $\beta_1 = 0.9$, $\beta_2 = 0.999$ and weight decay set to 0. We adopt cosine annealing with warmup start (Loshchilov & Hutter, 2016) to schedule the learning rate, which warms up to 2.5e-3 in the first 10% of total iterations and ends at 2.5e-4. The pretraining dataset is C4, the model max length is set to $L = 256$ and the token batch size is set to $N_B = 256 \times 512 = 131072$.

### A.3.3 Evaluation on computational time costs of different baselines and DeBLAS

Conceptually, the periodical evaluation on the calibration set of the Balanced Batching stage of DeBLAS incurs additional computation cost compared to regular training. In this section, we provide an empirical analysis of the run-time computational cost of DeBLAS compared with Random to examine the actual training efficiency of our method. We run DeBLAS with different model sizes on a single NVIDIA A100 GPU and measure the average duration of a regular training iteration and an adjustment of sampling probabilities in the Balanced Batching stage.

Table 12: Average time (in seconds) for a regular training iteration and an adjustment of sampling probabilities in the Balanced Batching stage of DeBLAS, evaluated on a single NVIDIA A100 GPU.

| Model Size | Training Iteration | Sampling Adjustment |
|---|---|---|
| 60M | 0.53 | 5.12 |
| 130M | 1.03 | 9.02 |
| 350M | 3.21 | 15.45 |
| 1B | 16.50 | 53.23 |

As shown in Table 12, the average time required for an evaluation on the calibration set

remains under 10 times greater than a regular training iteration. Notably, as model size scales from 60M to 1B, the relative overhead of the sampling probabilities adjustment decreases significantly, demonstrating the scalability of DeBLAS.

### A.3.4 Discussion on the scalability of DeBLAS regarding context length and token budget

**The impact of the length distribution of corpora.** The length distribution of the pretraining corpora is an important factor to consider when deciding the context window length and the data batching strategy. As illustrated in Figure 6, nearly half of the sequences in the C4 dataset contain less than 256 tokens and nearly half of the sequences in the SlimPajama dataset contain less than 512 tokens. Therefore, it's reasonable to select the context lengths as 256 and 512 for C4 and SlimPajama, respectively, to trade off between computation efficiency and data coverage under the padding mechanism. Setting the context length too short will waste most of the tokens in the corpora, while setting it too long will result in massive computational waste due to excessive padding tokens.

**Scaling to longer context and larger token budget.** In order to scale our method to longer context and larger token budget, we design an adapted version of DeBLAS that maximizes computation efficiency and data utilization simultaneously. We present the detailed implementation in Algorithm 2. Specifically, there are two major adaptations concerning the two stages, respectively.

**The Dense Batching stage.** Instead of using a fixed uniform length as in Algorithm 1, the dense batch length $L_d$ progressively increases to match the natural length distribution of the corpus. To be specific, the $L_d$ is increased over the range $[\frac{L}{K-1}, \frac{2L}{K-1}, \cdots, L]$, where $L$ is the model context length and $K$ is the number of length bins. When $L_d$ is $\frac{iL}{K-1}$, we only sample sequences from the length bin $[\frac{iL}{K-1}, \frac{(i+1)L}{K-1})$ to construct dense batches. We allocate training budget to a specific length for $L_d$, $\frac{iL}{K-1}$, proportional to the size of its corresponding sampling bin, $[\frac{iL}{K-1}, \frac{(i+1)L}{K-1})$.

**The Balanced Batching stage.** Instead of mixing data from different length bins in a single batch, Algorithm 2 samples a single length bin based on the calculated sampling probabilities, and constructs a batch exclusively with sequences from that specific bin. The procedure of calculating the sampling probabilities for each length bin is the same as Algorithm 1.

**Efficiency gain of Algorithm 2.** In the case of a large token budget, keeping $L_d$ fixed means choosing a relatively small $L_d$ because there might not be sufficient sequences longer than a large $L_d$ for the extended Dense Batching stage. This would cause significant token waste due to the truncation of long sequences. In Algorithm 2, only sequences from the bin $[\frac{iL}{K-1}, \frac{(i+1)L}{K-1})$, instead of all sequences longer than $\frac{iL}{K-1}$, are used to construct the dense batches with $L_d = \frac{iL}{K-1}$. This technique effectively reduces token waste due to the truncation of long sequences. What's more, the model is first trained on dense batches composed of shorter sequences and subsequently exposed to dense batches composed of increasingly longer sequences, eventually reaching the full context length before entering the Balanced Batching stage. Therefore, long-context sequences are already introduced during this stage. Furthermore, in the case of long-context modeling, constructing a padded batch with sequences from all possible length bins would result in massive padding tokens because of highly varied lengths within a long context window. The Balanced Sampling stage of Algorithm 2 effectively reduces the padding tokens by sampling from only one bin at each training iteration.

**Scaling experiment details.** We pretrain a LLaMA 130M model on SlimPajama with this adapted DeBLAS, using a token budget of 100B and a context length of 2048. We set the number of sequence in a batch $B_S$ to 1024 so the token batch size $N_B = 1024 \times 2048 = 2097152$ and the number of total training iterations is 50000. We set $T_d$ to 40% of the total training iterations, the number of length bins $K$ to 4, $N_C = 2000$ and $T_C = 2500$. As shown in Figure 4(c), DeBLAS can still achieve significant pretraining acceleration under practical industry-level pretraining conditions. We additionally measure the end-to-end runtime for the long-context experiment. Under the same hardware setting with NVIDIA A100 GPUs, Random and DeBLAS require 275.39 and 169.12 GPU hours, respectively, further demonstrating that the optimization iteration reduction achieved by DeBLAS translates effectively into practical runtime acceleration in long-context pretraining settings.

---

**Algorithm 2** DeBLAS for Long Context and Large Token Budget

---

**Require:** learning rate $\eta$, token batch size $N_B$, total iterations $T$, Dense Batching stage iterations $T_d$, model max length $L$, number of length bins $K$, calibration set size $N_C$, calibration frequency $T_C$

**Ensure:** pretrained model $\boldsymbol{\theta}_T$

1: Define length bins: $[0, \frac{L}{K-1}), [\frac{L}{K-1}, \frac{2L}{K-1}), \ldots, [\frac{(K-2)L}{K-1}, L), [L]$
2: Randomly sample $N_C$ sequences from the training set to construct a calibration set $D_C$ and compute the length ratio vector $\boldsymbol{r}_C = [r_1, r_2, \cdots, r_K]$
3: Compute the time step to increase the uniform length $L_d$: $[0, t_1, t_2, \cdots, t_i, \cdots, t_{K-1}]$ to align with the natural length distribution of the training set.
4: **for** $t = 1$ to $T_d$ **do**
5:     Find $i$ such that $t_{i-1} \le t < t_i$
6:     $L_d \leftarrow \frac{iL}{K-1}, \quad B_d \leftarrow \frac{N_B}{L_d}$
7:     Sample sequences from the length bin $[\frac{iL}{K-1}, \frac{(i+1)L}{K-1}]$ to construct a dense batch $\{\boldsymbol{x}^{(i)}\}_{i=1}^{B_d}$ of the uniform length $L_d$
8:     $\boldsymbol{\theta_t} \leftarrow \boldsymbol{\theta_{t-1}} - \eta \nabla_{\boldsymbol{\theta_{t-1}}} \left\{ \ell(\{\boldsymbol{x}^{(i)}\}_{i=1}^{B_d}, \boldsymbol{\theta}_{t-1}) \right\}$
9: **end for**
10: **for** $t = T_d + 1$ to $T$ **do**
11:     **if** $(t - T_d) \bmod T_C = 0$ **then**
12:         Evaluate $\theta_{t-1}$ on $D_C$, yielding losses on $K$ bins $\boldsymbol{l}_C = [l_1, l_2, \cdots, l_K]$
13:         $P_k \leftarrow \frac{r_k l_k}{\sum_{j=1}^{K} r_j l_j}$
14:     **end if**
15:     Sample $k \sim [P_1, \ldots, P_K]$
16:     $B \leftarrow \frac{N_B(K-1)}{kL}$
17:     Sample a padded batch $\{\boldsymbol{x}^{(i)}\}_{i=1}^{B}$ from the length bin $[\frac{(k-1)L}{k-1}, \frac{kL}{k-1}]$
18:     $\boldsymbol{\theta_t} \leftarrow \boldsymbol{\theta_{t-1}} - \eta \nabla_{\boldsymbol{\theta_{t-1}}} \left\{ \ell(\{\boldsymbol{x}^{(i)}\}_{i=1}^{B_d}, \boldsymbol{\theta}_{t-1}) \right\}$
19: **end for**

---

**Further evaluation on the 4K context length.** To further evaluate DeBLAS under modern long-context pretraining settings, we additionally conduct experiments using a context length of 4096. To better capture the broader sequence-length distribution under longer contexts, we increase the number of length bins $K$ to 5. The token batch size, total token budget, optimizer settings, and all other hyperparameters remain identical to those used in the 2048-context experiment. As shown in Table 13, DeBLAS continues to require substantially fewer optimization iterations than Random to reach the same target validation perplexity. These results suggest that the proposed dense-to-balanced scheduling strategy remains effective under long context windows.

Table 13: Evaluation on a context length of 4096. Training Iterations (x1000) (↓) to reach the given validation perplexity (PPL) and values of relative speedup against Random are reported.

| Target PPL | Method | Iterations | Speedup |
|---|---|---|---|
| 15.2 | Random | 50 | – |
| | DeBLAS | 31 | 1.61× |

### A.3.5 Comparison between DeBLAS and Dataset Decomposition (DD)

Dataset Decomposition (DD) (Pouransari et al., 2024) is a recent text data preprocessing method that decomposes text corpora based on sequence length and can be utilized to train language models with variable sequence length (VSL) and length-based curriculum. It has been demonstrated to improve downstream task performance and training efficiency for language model pretraining. Specifically, DD decomposes a given corpus containing documents of variable lengths into a collection of buckets, where every bucket $\mathcal{D}_i$ contains sequences of length $2^i$, each extracted from a unique document. During training with VSL, at every training iteration, a bucket index $i$ is sampled based on a specified curriculum to form a batch with $b/2^i$ sequences from the bucket $D_i$, keeping the total number of tokens in a batch constant ($2^i \times b/2^i = b$). DD eliminates the need

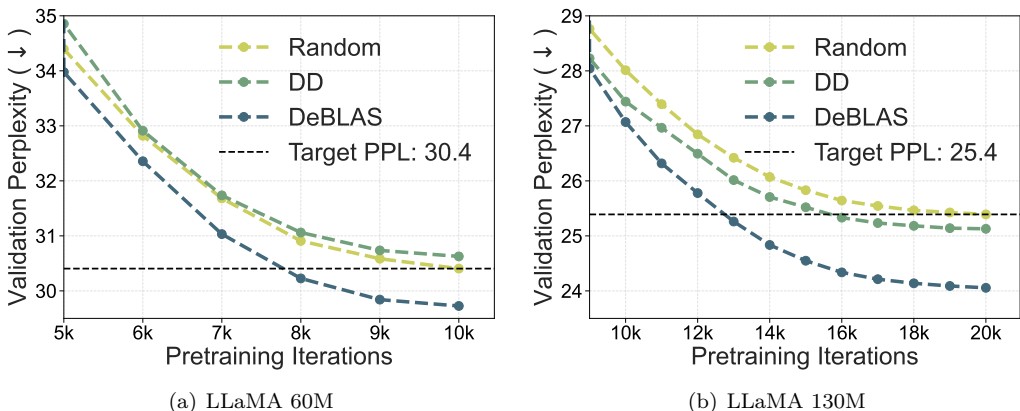

(a) LLaMA 60M                                        (b) LLaMA 130M

Figure 8: Validation perplexity over iterations of LLaMA 60M and 130M pretrained with DD and DeBLAS.

Table 14: Comparison between DD-Grow-P2, DeBLAS-VSL and DeBLAS. The training time is measure in hours and the required iterations (x1000) are in brackets.

| Model size | Target PPL | Method | | | |
|---|---|---|---|---|---|
| | | Random | Grow-P2 | DeBLAS | DeBLAS-VSL |
| 60M | 30.4 | 1.56 (10) | 1.69 (11) | 1.24 (8) | 1.23 (8) |
| 130M | 25.4 | 6.22 (20) | 5.58 (18) | 4.02 (13) | 3.70 (12) |

for padding tokens while making sure tokens in each sequence are from the same document by construction, which achieves maximum token utilization rate (TUR) in the context of autoregressive pretraining.

Conceptually, DD confines the model to be exposed to a limited number of length choices (e.g. $2^i$), which may cause length-wise bias. In contrast, DeBLAS enables training across all possible sequence lengths at the cost of padding tokens at the Balanced Batching stages, which mitigates the length bias induced by the Dense Batching stage. In addition, the sampling probabilities for different length choices in DD require manual tuning, restricting its scalability and practicality, while DeBLAS automatically determines the sampling probabilities based on the training loss statistics.

DD reduces training time cost of optimization steps in which short-sequence buckets are sampled, compared to training on sequences of model context length. However, in the Dense Batching stage of DeBLAS, the model is pretrained on dense batches with a uniform sequence length $L_d$ which can be significantly shorter than the model context length. To be specific, we set $L_d$ to half of the model context length in our main experiments, which means that a single optimization step of Dense Batching stage can be also faster than a standard training optimization step. Furthermore, since DD and DeBLAS both keep the total number of tokens in a batch constant, the idea of variable sequence length (VSL) in DD can be also applied to both stages of DeBLAS, further pushing the pretraining speed of DeBLAS. To simplify the design, we consider applying VSL only to the Balanced Batching stage as follows. When constructing a training data batch, instead of sampling from all the length bins, we sample from only one length bin, which we choose from all the length bins according to their sampling probabilities. We call this variant of DeBLAS as DeBLAS-VSL.

Empirically, we conduct an experiment to compare the pretraining efficiency of DD with the uniform curriculum and DeBLAS. As shown in Figure 8, DD shows no significant efficiency improvement over random selection, while our DeBLAS can achieve the target validation perplexity in much fewer iterations. To provide a more comprehensive comparison between DeBLAS and Dataset Decomposition, we pretrain LLaMA 60M and 130M models on C4 to reach the target validation perplexity in Table 1 of our manuscript with the Grow-P2 curriculum for Dataset Decomposition, which is reported as the optimal curriculum in the original paper. We compare it with DeBLAS and DeBLAS-VSL. We report the actual training time (in hours) and required training iterations of these methods when they reach the target perplexity on the validation set in Table 14. DeBLAS still outperforms Dataset Decomposition in terms of both metrics, and it can also benefit from the VSL technique.

Table 15: Evaluation on different domains of SlimPajama.

| Model size | Method | Pretraining Iterations | Commoncrawl | C4 | Github | Books | Arxiv | Wikipedia | StackExchange |
|---|---|---|---|---|---|---|---|---|---|
| 60M | Random | 20 | 28.73 | 31.91 | 5.51 | 30.63 | 19.72 | 17.10 | 12.95 |
| | DeBLAS | 14 | 28.64 | 32.17 | 5.59 | 29.68 | 19.72 | 17.23 | 13.09 |
| 130M | Random | 40 | 23.42 | 26.40 | 4.64 | 23.48 | 16.48 | 13.37 | 10.84 |
| | DeBLAS | 28 | 23.05 | 26.39 | 4.61 | 22.76 | 16.27 | 13.29 | 10.76 |
| 350M | Random | 60 | 18.88 | 21.57 | 3.91 | 17.73 | 13.43 | 10.17 | 8.99 |
| | DeBLAS | 39 | 18.68 | 21.74 | 3.92 | 16.91 | 13.29 | 10.26 | 9.00 |
| 1B | Random | 100 | 16.50 | 19.02 | 3.52 | 14.24 | 11.81 | 8.58 | 8.03 |
| | DeBLAS | 62.5 | 16.44 | 19.04 | 3.57 | 13.95 | 11.81 | 8.46 | 8.10 |

### A.3.6 Comparison with reference-model-based methods

To empirically compare reference-model-based methods against reference-model-free methods, we conduct experiments on two representative reference-model-based baselines, RHO-LOSS (Mindermann et al., 2022) and Bayesian Data Selection (BDS) (Deng et al., 2023). Both methods select a data batch with the same batch size as regular training from a larger batch at every iteration. RHO-LOSS selects training data that maximize the difference between the training loss of the current model and an "irreducible loss" estimated by an additional reference model trained on a holdout dataset. BDS chooses training data based on a Bayesian estimate of their influence on generalization loss, using a lightweight Laplace approximation together with a pretrained model. In our experiment, we use the LLaMA 60M model pretrained with the Random baseline, which is presented in Table 1, as the reference model for these two methods. We set the large candidate batch size to be twice the base sequence batch size.

We pretrain LLaMA 130M models on C4 and measure the number of iterations required to reach the target validation perplexity used in Table 1, as well as the average per-iteration training time and total training time. As presented in Table 4, our DeBLAS still requires substantially fewer training iterations than these two reference-model-based baselines. Furthermore, RHO-LOSS and BDS incur considerable time overhead at every iteration because both the reference model and the current model must perform forward passes on all candidate sequences, the number of which is twice that of selected sequences. Consequently, RHO-LOSS and BDS require significantly more wall-clock training time than even the Random baseline to reach the same validation perplexity, demonstrating the efficiency gain of reference-model-free methods. Since we set the sequence length $L_d$ in the Dense Batching stage to be half of the model context length, the average per-iteration training time of DeBLAS is slightly less than Random due to the quadratic complexity of the attention mechanism.

### A.3.7 Memory efficiency

We evaluate the practical GPU memory efficiency of DeBLAS by profiling GPU utilization and peak GPU memory consumption on a single NVIDIA A100 GPU with LLaMA 130M model and C4 dataset in Table 5. The experimental setup follows the configuration used in the main experiments of Table 1. We separately profile the Random baseline, the Dense Batching stage, and the Balanced Batching stage of DeBLAS. For GPU utilization, we monitor the streaming multiprocessor (SM) utilization with a sampling interval of 1 second. Each method is profiled over 8000 consecutive training iterations, and the first 100 iterations are discarded as warmup for stabilization. The reported GPU utilization is computed as the average SM utilization over the remaining stable iterations.

### A.3.8 Stability Analysis of Learned Sampling Probabilities

To evaluate whether the calibration process of the Balanced Batching stage produces stable sampling behaviors, we conduct a stability analysis across random seeds, datasets, and model scales. Specifically, we train LLaMA models of sizes 60M, 130M, and 350M on both C4 and SlimPajama using three random seeds. For each configuration, we record the learned sampling probabilities of every length bin at each calibration

step throughout the Balanced Batching stage. Let $p_{k,t}^{(s)}$ denote the sampling probability of length bin $k$ at calibration step $t$ under random seed $s$. For each length bin and calibration step, we first compute the mean probability across seeds:

$$\bar{p}_{k,t} = \frac{1}{S} \sum_{s=1}^{S} p_{k,t}^{(s)}, \tag{7}$$

where $S$ is the number of random seeds. The standard deviation across seeds is then:

$$\sigma_{k,t} = \sqrt{\frac{1}{S} \sum_{s=1}^{S} \left( p_{k,t}^{(s)} - \bar{p}_{k,t} \right)^2}. \tag{8}$$

To summarize the overall stability of a training run, we average the standard deviation over all $K$ length bins and $T$ calibration steps:

$$\text{AvgStd} = \frac{1}{KT} \sum_{t=1}^{T} \sum_{k=1}^{K} \sigma_{k,t}. \tag{9}$$

Table 6 reports AvgStd for all evaluated settings.

### A.3.9 Discussion on other data attributes

Our work focuses on sequence length for pretraining acceleration because it is a quantitative, easily measurable factor that directly affects token-level utilization and thus plays a central role in pretraining efficiency. In contrast, semantic difficulty lacks a reliable and universally applicable metric, and detailed domain annotations are often unavailable or prohibitively expensive to obtain at scale, which makes it difficult to incorporate such attributes into a controllable data scheduling framework.

Nonetheless, DeBLAS already implicitly accounts for semantic difficulty. During the Balanced Batching stage, length bins are reweighted based on their evaluation losses, which serve as a commonly used proxy for semantic difficulty in practice. As a result, more semantically challenging data naturally receives higher weight, even without explicit difficulty annotations.

Regarding domain distribution, the pretraining corpora we used in the main experiments, i.e., C4 and SlimPajama, already reflect highly diverse and complex real-world data distributions. To be more specific, C4 is a cleaned version of Common Crawl's web crawl corpus and SlimPajama contains extensive deduplicated and curated data drawn from multiple domains including CommonCrawl, C4, GitHub, Books, arXiv, Wikipedia, and StackExchange. As shown in Table 1 and 2 in our paper, DeBLAS outperforms all the baselines on both C4 and SlimPajama, which can demonstrate the robustness of DeBLAS under real-world complex data scenarios.

To further validate the generalization capabilities of DeBLAS, we evaluate models pretrained with Random and DeBLAS on the validation data of different domains in SlimPajama. As shown in Table 15, DeBLAS achieves comparable or lower perplexity than the Random baseline across different domains, indicating that our method generalizes well in the presence of diverse semantic distributions.

### A.3.10 Evaluation results on multiple trials

We conduct multiple training runs of pretraining LLaMA 60M and 130M models on C4 with Random and DeBLAS and report the results with standard deviation in Figure 9. As illustrated, DeBLAS can consistently outperform Random in terms of the reduction of iterations required to reach the same level of validation perplexity.

### A.3.11 Performance of DeBLAS on fine-grained length bins

Under the training framework of DeBLAS, the Balanced Batching stage is designed to mitigate the length-wise bias induced by the Dense Batching stage. To validate its necessity, we present the validation perplexity progression of DeBLAS with varying uniform lengths of the Dense Batching stage, comparing it with the

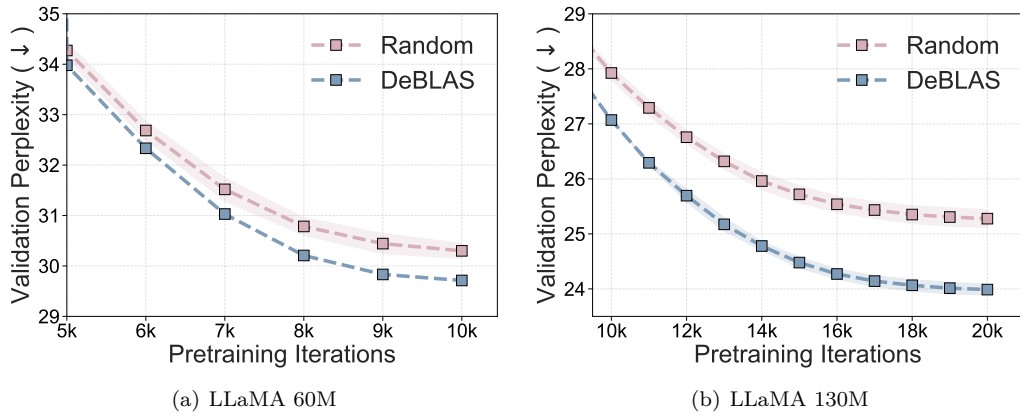

(a) LLaMA 60M      (b) LLaMA 130M

Figure 9: Validation perplexity over pretraining iterations of LLaMA 60M and 130M with standard deviation based on multiple training runs on the C4 dataset.

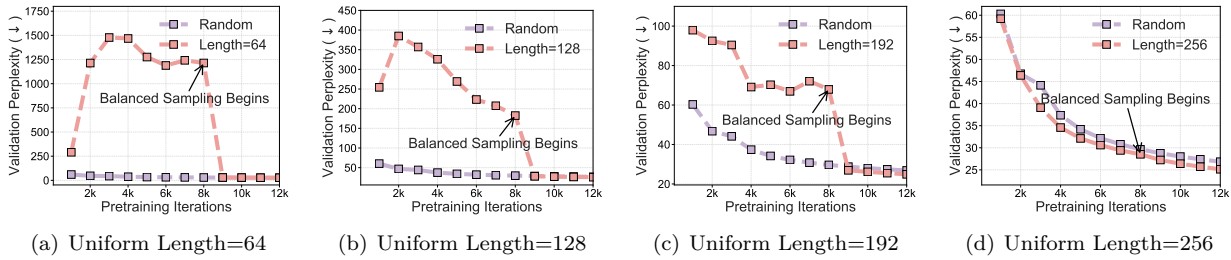

(a) Uniform Length=64    (b) Uniform Length=128    (c) Uniform Length=192    (d) Uniform Length=256

Figure 10: Validation perplexity over pretraining iterations of LLaMA 130M pretrained on C4 with DeBLAS using different uniform lengths of the Dense Batching stage.

Random baseline, in Figure 10. For uniform lengths less than the model max length, i.e., 256, the model exhibits severe performance degradation compared with Random in terms of the overall perplexity on the validation set of C4, which consists of sequences of diverse length ranges. After the Balanced Batching stage begins, the model exhibits a rapid decline in perplexity and eventually surpasses Random, which demonstrates its effectiveness in alleviating the length-wise bias. Notably, as illustrated in Figure 10(d), training LLaMA 130M models with C4 exclusively on dense batches of uniform length=256 achieves lower validation perplexity than Random, aligning with the result in Table 1. However, finer-grained evaluation across different length bins reveals a trade-off: we measure the model performance on fine-grained length bins and find that the acceleration effect achieved by dense batches of uniform length=256 comes at the cost of worse language modeling on shorter sequences. In contrast, DeBLAS achieves robust perplexity improvements across all length bins, balancing efficiency with comprehensive generalization.

### A.3.12    Detailed results on scaling to LLaMA 7B

We present the experiment results of a comparison between Random and DeBLAS on LLaMA 7B model pretrained with Slimpajama in Figure 11(b). For DeBLAS, we set the iterations for the Dense Batching stage to 6000. Since we evaluate the model every 1000 iterations and the Dense Batching stage induces significant length-wise bias, we report the validation perplexity of DeBLAS and Random starting from the 7000th iterations. As shown in Figure 11(b), DeBLAS reduces the required iterations significantly to reach the same level of validation perplexity as Random, which demonstrates that our method can significantly accelerate LLaMA 7B pretraining compared with Random.

### A.3.13    Experiments on replacing the dense batch with the packed batch

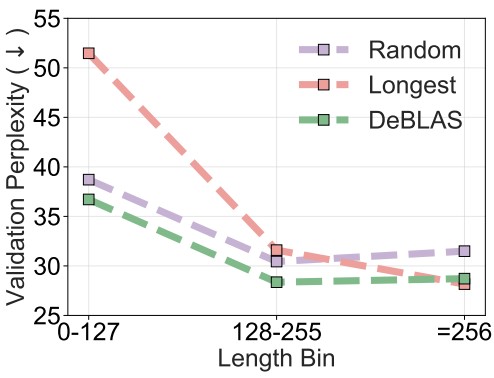 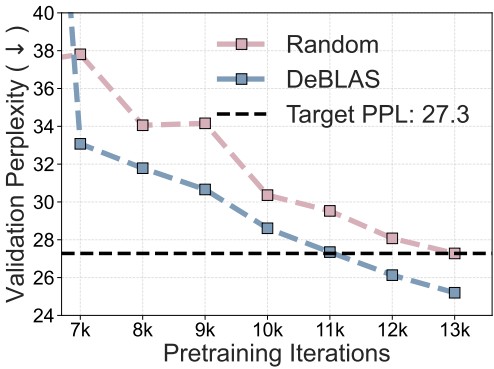

(a) Comparison on different length bins in LLaMA 130M with C4

(b) Comparison on LLaMA 7B pretrained with Slimpajama

Figure 11: (a) Comparison on length bins in LLaMA 130M with C4. Longest in the figure represents training exclusively on dense batches of uniform length equal to the model max length, i.e., 256. We pretrained three LLaMA 130M models with Random, Longest and DeBLAS, respectively, and measure their perplexity on validation data divided into the three length bins; (b) Comparison between Random and DeBLAS on LLaMA 7B pretrained with Slimpajama. DeBLAS outperforms Random in terms of reduced iterations required to reach the same level of validation perplexity.

Table 16: Evaluation on long-context understanding benchmarks.

| Benchmark | 2WikiMultihopQA | SQuAD-3-shots | ArcEasy-3-shots | ArcChallenge-3-shots | MMLU-5-shots |
|---|---|---|---|---|---|
| Random | 0.80 | 50.07 | 30.81 | 24.06 | 22.95 |
| DeBLAS | 1.70 | 50.07 | 31.06 | 25.09 | 22.95 |

In order to investigate whether maximized token utilization is the primary factor behind the pretraining acceleration observed with dense batching, we replace the dense batch in the Dense Batching stage with the packed batch, which is constructed by the packing mechanism. We pretrain a LLaMA 130M model with C4 with this method and compare it with DeBLAS. The packed batch is filled with semantically meaningful tokens, thereby eliminating the explicit waste of padding tokens. But the packed batch can also suffer from token utilization loss when complete documents are split into different segments. As shown in the Figure 12, replacing the dense batch with the packed batch significantly slows down the pretraining, measured by the required training iterations to achieve the same validation perplexity, which demonstrates that the token utilization maximization of the dense batch is indeed the key factor of its acceleration effect.

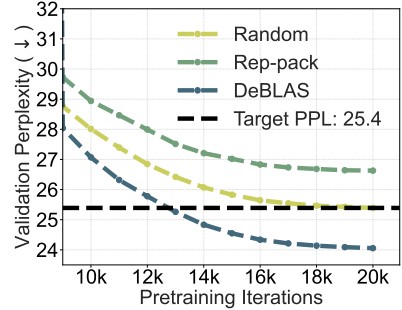

Figure 12: Experiments on replacing dense batches with packed batches. Rep-pack denotes replacing with packed batches.

### A.3.14 Discussion on long-context benchmarks.

To evaluate the impact of our method on long-context understanding, we conducted an additional experiment by pretraining two LLaMA-130M models on C4 using DeBLAS and random sampling, respectively. Both models were trained with a maximum context length of 1024 and a token budget of 10B. We set the context length to 1024 since over 90% of the C4 sequences are shorter than 1024. We then evaluated them on five new benchmarks, including one long-context comprehension task (2WikiMultihopQA (Bai et al., 2024)) and four few-shot evaluations on standard benchmarks (Pouransari et al., 2024). Due to the length distribution disparity between C4 and long-context downstream tasks, the models exhibit poor performance on the 2WikiMultihopQA. Therefore, we incorporate 4 few-shot evaluations of normal-length benchmarks

following (Pouransari et al., 2024). As shown in Table 16, DeBLAS enables the model to reach the same level of performance in fewer iterations than Random across all evaluated benchmarks.

## B    Discussion of Related Works

### B.1    Related Work

**Online data selection.**    Online data selection speeds up model training by dynamically selecting training samples in every iteration. (Loshchilov & Hutter, 2015; Jiang et al., 2019; Katharopoulos & Fleuret, 2018) scheduling hard samples based on simple heuristics like training loss or gradient norm, which can't be guaranteed to capture the true informativeness of the training data and can be sensitive to outliers (Hong et al., 2024). Consequently, these methods have been demonstrated to fall short in performance compared to random selection in some cases. Another line of work achieves notable acceleration by leveraging additional reference models to select valuable training samples (Deng et al., 2023; Mindermann et al., 2022). However, their applications in practice are constrained by the availability and non-negligible computational cost of well-performing reference models (Kaddour et al., 2023).

**Data selection for LLM pretraining.**    Data selection is pivotal in optimizing the efficiency and performance of LLM pretraining (Albalak et al., 2022; Wan et al., 2023). Offline data selection has shown notable success in LLM pretraining by selecting high-quality data from extensive web corpora. Heuristic approaches apply rule-based filters, such as removing short, repetitive, and toxic contents (Raffel et al., 2020; Rae et al., 2021; Brown et al., 2020). Reference-model-based methods leverage models trained on trusted corpora (e.g., Wikipedia and Books) or publicly available pretrained models to assign a utility score to candidate data (Brown et al., 2020; Du et al., 2022; Wenzek et al., 2019; Xie et al., 2023b; Wettig et al., 2024). In contrast, the application of online data selection for language model pretraining remains underexplored. GREATS (Wang et al., 2024) leverages Taylor expansions to approximate the influence of training data on validation loss and designs a "ghost inner-product" technique to further reduce the actual runtime. But its evaluation on pretraining is limited to a small-scale setting. MATES (Yu et al., 2024) trains a light-weight proxy model to approximate the influence score on the full training corpora. However, the limited capacity of the proxy model compromises the accuracy of the estimation of influence scores (Zhang et al., 2024a).

**Impact of sequence length.**    The sequence length used during the LLM pretraining plays a crucial part in determining both the computational demands and the textual representations captured by the model (Pouransari et al., 2024; Li et al., 2022). To be specific, it refers to the number of tokens contained in the tokenization, which directly impacts computational efficiency due to the quadratic complexity of the attention mechanism (Vaswani et al., 2017). (Variš & Bojar, 2021) shows that the performance of the Transformer model declines notably when handling sequences whose lengths deviate from the distribution of lengths present in the training data on a string editing task. (Anil et al., 2022) points out that LLMs struggles to extrapolate from short problem instances to longer ones in reasoning tasks after naive fine-tuning. (Pouransari et al., 2024) establishes that the distribution and curriculum of sequence lengths during training lead to a significant impact on the performance of large language models in various natural language and long-context understanding benchmarks. Sequence Length Warmup (Li et al., 2022) gradually increase the sequence length during pretraining to improve stability.

**Length-based curriculum.**    Attempts have been made to improve pretraining efficiency by controlling the sequence length distribution during training. GrowLength (Jin et al., 2023) progressively increase the sequence length according to a predefined curriculum during the training phase to speed up convergence. Dataset Decomposition (DD) (Pouransari et al., 2024) partitions the training corpus into a series of buckets, where every bucket $D_i$ contains sequences of length $2^i$, and a bucket is sampled based on a pre-defined curriculum at every training iteration. Technically, DeBLAS differs from previous length-based curriculum methods in two important aspects. Firstly, DeBLAS employs an adaptive second stage that dynamically adjusts length-bin sampling probabilities according to online loss statistics collected throughout training. Secondly, previous methods neglect the length-wise bias potentially induced by restricting all sequences in a batch to a narrow length range. In contrast, DeBLAS exposes the model to all sequence lengths

during training and adaptively reallocates training budget toward underrepresented length bins through the Balanced Batching stage, thereby mitigating the bias introduced by the Dense Batching stage.

## C   Theoretical Analysis about Convergence Rate

In this section, we formally present the full theoretical analysis with detailed definitions and assumptions. Based on the insights in Section 3.1, we can define the objective of Dense-Balanced Sequence Scheduling for LLM pretraining as follows:

$$\min_{\theta} \mathcal{L}_{\text{curr}}(\theta) = \mathbb{E}_{x \sim \pi_t(x)} [\ell(x; \theta)] = \mathbb{E}_{x \sim \pi_t(x)} \left[ -\sum_{i=1}^{|x|} \log P_\theta(x_i \mid x_{<i}) \right], \tag{10}$$

where $\pi_t(x)$ is a time-dependent (e.g., training iterations) scheduling sampling distribution as:

$$\pi_t(x) = \frac{s_t(x)}{\sum_{x' \in \mathcal{D}} s_t(x')}, \quad s_t(x) = (1 - \lambda_t) \cdot |x| + \lambda_t \cdot \ell(x; \theta_t), \tag{11}$$

where $s_t(x)$ denotes a curriculum score, $|x|$ denotes the length (in tokens) of sample $x$ (preprocessed with the padding mechanism), $\ell(x; \theta_t)$ is the current model loss on $x$, and $\lambda_t \in [0, 1]$ is a curriculum scheduler over time (e.g., stage-wise). This schedule prioritizes long-context inputs to construct dense batches for maximization of token utilization in early training stages ($\lambda_t \to 0$), and subsequently shifts to high-loss inputs to correct length-wise bias for balanced learning performance across varied length ranges ($\lambda_t \to 1$).

Let $\mathcal{L}(\theta) = \mathbb{E}_{x \sim \mathcal{D}}[\ell(x; \theta)]$ be general training objective, where $\ell(x; \theta)$ is the token-level cross-entropy loss over sequence $x = (x_1, \cdots, x_{|x|})$, and $\theta$ are the model parameters. We define a two-stage curriculum: **Stage 1:** sample $x \sim \pi_t(x) \propto |x|$ (favoring longer sequences); **Stage 2:** sample $x \sim \pi_t(x) \propto \ell(x; \theta_t)$ (favoring higher-loss sequences).

*Proof of Lemma 2.* Let $f(\boldsymbol{x}) = \|\nabla \ell(\boldsymbol{x}; \theta)\|^2$ denote the squared gradient norm and $h(\boldsymbol{x}) = \ell(\boldsymbol{x}; \theta)$ denote the loss. In the **Balanced Batching** stage, $\pi(\boldsymbol{x}) \propto h(\boldsymbol{x})$. The Radon-Nikodym derivative of $\pi$ w.r.t. $\mathcal{D}$ is:

$$\frac{d\pi}{d\mathcal{D}}(\boldsymbol{x}) = \frac{h(\boldsymbol{x})}{\mathbb{E}_{z \sim \mathcal{D}}[h(z)]} \tag{12}$$

We aim to prove $\mathbb{E}_{\boldsymbol{x} \sim \pi}[f(\boldsymbol{x})] \geq \mathbb{E}_{\boldsymbol{x} \sim \mathcal{D}}[f(\boldsymbol{x})]$. Expanding the expectation:

$$\mathbb{E}_{\boldsymbol{x} \sim \pi}[f(\boldsymbol{x})] = \int f(\boldsymbol{x}) \frac{h(\boldsymbol{x})}{\mathbb{E}_{z \sim \mathcal{D}}[h(z)]} d\mathcal{D}(\boldsymbol{x}) \tag{13}$$

$$= \frac{1}{\mathbb{E}_{\boldsymbol{x} \sim \mathcal{D}}[h(\boldsymbol{x})]} \mathbb{E}_{\boldsymbol{x} \sim \mathcal{D}}[f(\boldsymbol{x}) \cdot h(\boldsymbol{x})] \tag{14}$$

The condition $\mathbb{E}_{\boldsymbol{x} \sim \pi}[f(\boldsymbol{x})] \geq \mathbb{E}_{\boldsymbol{x} \sim \mathcal{D}}[f(\boldsymbol{x})]$ is equivalent to:

$$\frac{\mathbb{E}_{\boldsymbol{x} \sim \mathcal{D}}[f(\boldsymbol{x})h(\boldsymbol{x})]}{\mathbb{E}_{\boldsymbol{x} \sim \mathcal{D}}[h(\boldsymbol{x})]} \geq \mathbb{E}_{\boldsymbol{x} \sim \mathcal{D}}[f(\boldsymbol{x})] \tag{15}$$

Assuming $\mathbb{E}_{\mathcal{D}}[h(\boldsymbol{x})] > 0$, we rearrange to find the definition of covariance:

$$\mathbb{E}_{\boldsymbol{x} \sim \mathcal{D}}[f(\boldsymbol{x})h(\boldsymbol{x})] - \mathbb{E}_{\boldsymbol{x} \sim \mathcal{D}}[f(\boldsymbol{x})]\mathbb{E}_{\boldsymbol{x} \sim \mathcal{D}}[h(\boldsymbol{x})] \geq 0 \iff \text{Cov}_{\mathcal{D}}(f(\boldsymbol{x}), h(\boldsymbol{x})) \geq 0 \tag{16}$$

Therefore, the inequality $E_\pi \geq E_D$ is sufficiently proved by the positive covariance between the loss and gradient norm, which is exactly the Assumption 2, empirically supported by the experiment results in Figure 3(c). □

The foundational theory for optimization in non-convex settings (Carmon et al., 2018; Nesterov et al., 2018) and practical approaches (Graves et al., 2017; Raj et al., 2020) for improving learning efficiency through curriculum and importance sampling techniques laid analytical foundations to the two-stage curriculum learning framework proposed in this analysis.

*Proof of Theorem 1.* We follow the standard analysis for SGD on non-convex $L$-smooth objectives, as described in (Carmon et al., 2018). At each step $t$, we update:

$$\theta_{t+1} = \theta_t - \eta \nabla \ell(x_t; \theta_t),$$

where $x_t \sim \pi_t$ is sampled from the curriculum distribution.

By the $L$-smoothness of $\mathcal{L}$, as discussed in (Nesterov et al., 2018), we have:

$$\mathcal{L}(\theta_{t+1}) \leq \mathcal{L}(\theta_t) + \langle \nabla \mathcal{L}(\theta_t), \theta_{t+1} - \theta_t \rangle + \frac{L}{2} \|\theta_{t+1} - \theta_t\|^2$$

$$= \mathcal{L}(\theta_t) - \eta \langle \nabla \mathcal{L}(\theta_t), \nabla \ell(x_t; \theta_t) \rangle + \frac{L\eta^2}{2} \|\nabla \ell(x_t; \theta_t)\|^2$$

Taking expectation over $x_t \sim \pi_t$:

$$\mathbb{E}[\mathcal{L}(\theta_{t+1})] \leq \mathbb{E}[\mathcal{L}(\theta_t)] - \eta \mathbb{E}[\|\nabla \mathcal{L}(\theta_t)\|^2] + \frac{L\eta^2}{2} \mathbb{E}_{x_t \sim \pi_t}[\|\nabla \ell(x_t; \theta_t)\|^2]$$

Now define:

$$\sigma_t^2 := \mathbb{E}_{x_t \sim \pi_t}[\|\nabla \ell(x_t; \theta_t) - \nabla \mathcal{L}(\theta_t)\|^2] \Rightarrow \mathbb{E}[\|\nabla \ell(x_t; \theta_t)\|^2] = \|\nabla \mathcal{L}(\theta_t)\|^2 + \sigma_t^2$$

Substitute into the above:

$$\mathbb{E}[\mathcal{L}(\theta_{t+1})] \leq \mathbb{E}[\mathcal{L}(\theta_t)] - \left( \eta - \frac{L\eta^2}{2} \right) \mathbb{E}[\|\nabla \mathcal{L}(\theta_t)\|^2] + \frac{L\eta^2}{2} \sigma_t^2$$

Let $\gamma = \eta - \frac{L\eta^2}{2} > 0$ (assumed by choosing small enough $\eta$), we get:

$$\mathbb{E}[\mathcal{L}(\theta_{t+1})] \leq \mathbb{E}[\mathcal{L}(\theta_t)] - \gamma \mathbb{E}[\|\nabla \mathcal{L}(\theta_t)\|^2] + \frac{L\eta^2}{2} \sigma_t^2$$

Sum from $t = 0$ to $T - 1$:

$$\mathbb{E}[\mathcal{L}(\theta_0)] - \mathbb{E}[\mathcal{L}(\theta_T)] \geq \gamma \sum_{t=0}^{T-1} \mathbb{E}[\|\nabla \mathcal{L}(\theta_t)\|^2] - \frac{L\eta^2}{2} \sum_{t=0}^{T-1} \sigma_t^2$$

Divide both sides by $T$:

$$\frac{1}{T} \sum_{t=0}^{T-1} \mathbb{E}[\|\nabla \mathcal{L}(\theta_t)\|^2] \leq \frac{\mathcal{L}(\theta_0) - \mathcal{L}^*}{\gamma T} + \frac{L\eta^2}{2\gamma T} \sum_{t=0}^{T-1} \sigma_t^2$$

Let $\sigma_t^2$ be decomposed as:

$$\sigma_t^2 = \begin{cases} \sigma_u^2 - \Delta\sigma_{\text{length}}^2, & \text{if } t < T_1 \\ \sigma_u^2 - \Delta\sigma_{\text{loss}}^2, & \text{if } t \geq T_1 \end{cases}$$

Then:

$$\sum_{t=0}^{T-1} \sigma_t^2 = T\sigma_u^2 - (T_1 \Delta\sigma_{\text{length}}^2 + T_2 \Delta\sigma_{\text{loss}}^2)$$

So the convergence bound becomes:

$$\frac{1}{T} \sum_{t=0}^{T-1} \mathbb{E}[\|\nabla \mathcal{L}(\theta_t)\|^2] \leq \mathcal{O}\left( \frac{1}{\sqrt{T}} \right) - \eta \cdot \left( \Delta\sigma_{\text{length}}^2 + \Delta\sigma_{\text{loss}}^2 \right)$$

as claimed.

$\square$

