# OpenReview forum: "DeBLAS: Accelerate LLM Pretraining by Length-based Sequence Scheduling"
_TMLR — Under review for TMLR_

### Review · Reviewer_VFTT · 2026-05-15

**Summary Of Contributions:**

This paper proposes a new data scheduling strategy for accelerating large language model pretraining.
In the first stage (Dense), the model trains on uniform-length dense batches to maximize token utilization and reduce gradient variance.
In the second stage (Padded), the method reweighs loss according to the sequence length to mitigate length-wise bias induced by the first stage.
Results suggest substantial reductions in required pretraining iterations while maintaining perplexity relative to standard random sampling.



### Strengths

1. The problem is clearly defined and well-motivated.
2. The methodology is conceptually simple and computationally lightweight. The method can be easily adapted into some existing training pipelines with low engineering overhead
3. The empirical gains in the number of iterations are high, which shows high practical relevance.


### Weaknesses

1. Some claims appear to be never supported by results.
2. No discussion or guidelines on selecting hyperparameters for a practical application.

**Audience:**

Yes

**Audience Explanation:**

A lightweight pretraining methodology is a solid contribution. It will be interesting for both research and engineering types of projects.

**Broader Impact Concerns:**

While the paper provides a Broader Impact Statement, this work presents a general pretraining methodology that brings no ethical concerns by itself.

**Claims And Evidence:**

No

**Claims Explanation:**

The paper presents a lot of different claims in the text, but only some of them are supported.

1. $\text{&#x2705;}$ "we provide a formal analysis of DeBLAS to demonstrate that it can accelerate the convergence of language model pretraining through the gradient variance reduction achieved by the combination of its Dense Batching and Balanced Batching stage (in Section 3.3)" This claim supported by a formal analysis is  Section 3.3 and Appendix C.
2. $\text{&#x2705;}$ "we conduct extensive experiments to verify the effectiveness of DeBLAS on two widely used pretraining corpora, i.e., C4 and SlimPajama, and perform various ablation studies and further discussions to provide a thorough understanding (in Section 4)" The manuscript contain extensive experiments and ablation study. This claim is supported by experiments in Section 4.
3. $\text{&#x274C;}$ "We empirically demonstrate that packing can actually reduce training efficiency compared to padding under a fixed token batch size (refer to Appendix A.3.1)." This claim is not supported by the results.
4. $\text{&#x274C;}$ "We prove that the model internalizes the foundational language knowledge during the dense-batch phase, allowing it to optimize more efficiently on the latter variable length sequences." This claim is not supported.
5. $\text{&#x2705;}$ "Empirical results show that our approach achieves comparable perplexity to standard pretraining with substantially fewer optimization steps, pinpointing a promising way to reduce the computational burden of LLM pretraining." This claim is fully supported by experiments in Section 4.
6. $\text{&#x274C;}$ "While these datasets may contain biases present in their source web-crawled data, our work does not amplify these biases and focuses on the scheduling of data rather than its content." This is a strong claim that is not supported by the content.

**Requested Changes:**

### Critical

The end of Section 2 introduces token utilization rate. It states, "We empirically demonstrate that packing can actually reduce training efficiency compared to padding under a fixed token batch size (refer to Appendix A.3.1)."
None of the tables or figures in the manuscript, including appendixes, ever demonstrate that.
Neither result seems to use a token utilization rate.
Why does this metric need to be introduced?
Could you show that "packing can actually reduce training efficiency compared to padding" ?

Generally, there are several claims in the text that are not supported by any results.
I encourage authors to go through the text and make sure that if it is written as "We prove...", "We demonstrate...", or "our work does...", there are results that prove, demonstrate, and do.
For example, which specific result is showing that the model "internalizes the foundational language knowledge during the dense-batch phase"? And what does "internalizes the foundational language knowledge" even mean? Or how did you test for biases to claim that they are not amplified?
Less vague statements would improve scientific rigor.

Fig. 3c is information dense and has an unfortunate colour scheme. From the description (including the appendix), I cannot understand what it shows. Consider recolouring one of the light greens. Besides that, it is unclear what "short-grad" or "short-loss" means. For example, "short-loss" never appears elsewhere in the text.
Also, please specify that the details of this figure are elaborated in Appendix A.3.2.

The experiments are reporting a decrease in the number of iterations. Table 4 also reports a wall-clock training time.
These are great results.
However, for the practical engineering applications, the memory-related metrics are also representative.
Could you report and compare to the random sampling:
1. GPU utilization for both stages.
2. Peak GPU memory utilization for both stages.

The paper presents several additional hyperparameters: base batch size, number of dense batching iterations, number of bins, uniform length of dense batch, and calibration set size.
There is no instruction on how to choose them if the methodology is applied for pretraining.
Could you discuss a set general guidelines on how to select those parameters?


### Minor

Table 4. Please specify that a number of iterations are reported in thousands. Otherwise, the math does not make sense.

The Broader Impact Statement is generally for any **direct** ethical implications of your work. Please do not treat it as a second conclusion. NeurIPS Guidlines(https://neurips.cc/public/guides/PaperChecklist) have a good discussion on what to include in such a statement. To quote it: "if you develop a generic algorithm for optimizing neural networks, you do not need to mention that this could enable people to train models that generate Deepfakes faster."

---

> ### Author Response · Authors · 2026-06-01
>
> ### On the weakness about unsupported claims
> > "We empirically demonstrate that packing can actually reduce training efficiency compared to padding under a fixed token batch size (refer to Appendix A.3.1)." This claim is not supported by the results.
>
> Thank you for the constructive feedback. Our intention is to claim that packing may require more optimization iterations than padding for the model to reach the same validation perplexity under a fixed token batch size, as observed in Figure 7 of Appendix A.3.1. To improve clarity, we have weakened the original statement to avoid overclaiming in the revised version: “Empirically, in Appendix A.3.1, we observe that packing requires more optimization iterations than padding for the model to reach the same validation perplexity under a fixed token batch size.” (refer to Section 2 of the revised manuscript)
>
> > "We prove that the model internalizes the foundational language knowledge during the dense-batch phase, allowing it to optimize more efficiently on the latter variable length sequences." This claim is not supported.
>
> Thank you for the valuable comment. We acknowledge that our original theoretical and empirical analyses do not formally prove that the model “internalizes foundational language knowledge.” More precisely, our empirical evidence shows that training with dense batches leads to lower gradient variance (Figure 3(b) and Lemma 1) and enables more efficient subsequent optimization on variable-length sequences (Figure 3(a) and Figure 5(b)). The original statement was intended as an intuitive interpretation of the observed optimization behavior.
>
> In the revised version, we have therefore replaced the original statement with a more precise one: “Our idealized theoretical analysis suggests that our approach can accelerate language model pretraining through the gradient variance reduction achieved by the combination of both stages.” (refer to Abstract)
>
> > "While these datasets may contain biases present in their source web-crawled data, our work does not amplify these biases and focuses on the scheduling of data rather than its content." This is a strong claim that is not supported by the content.
>
> Thank you for the constructive comment. We agree that the original statement is stronger than what is supported by the current paper. Our intention was to convey that DeBLAS operates on sequence scheduling rather than explicit semantic filtering or content modification. In addition, we would like to mention that Table 15 in the revised manuscript (also shown below) provides empirical evidence that models pretrained with DeBLAS achieve comparable perplexities across multiple SlimPajama domains against the Random baseline, indicating that DeBLAS maintains broadly consistent generalization behavior across diverse domains rather than over-optimizing toward particular subsets of the corpus.
>
> **Table. Evaluation on different domains of SlimPajama.**
> | Model Size | Method | Pretraining Iterations(K) | CommonCrawl | C4 | Github | Books | Arxiv | Wikipedia | StackExchange |
> |---|---|---|---|---|---|---|---|---|---|
> | 60M | Random | 20 | 28.73 | 31.91 | 5.51 | 30.63 | 19.72 | 17.10 | 12.95 |
> | 60M | DeBLAS | 14 | 28.64 | 32.17 | 5.59 | 29.68 | 19.72 | 17.23 | 13.09 |
> | 130M | Random | 40 | 23.42 | 26.40 | 4.64 | 23.48 | 16.48 | 13.37 | 10.84 |
> | 130M | DeBLAS | 28 | 23.05 | 26.39 | 4.61 | 22.76 | 16.27 | 13.29 | 10.76 |
> | 350M | Random | 60 | 18.88 | 21.57 | 3.91 | 17.73 | 13.43 | 10.17 | 8.99 |
> | 350M | DeBLAS | 39 | 18.68 | 21.74 | 3.92 | 16.91 | 13.29 | 10.26 | 9.00 |
> | 1B | Random | 100 | 16.50 | 19.02 | 3.52 | 14.24 | 11.81 | 8.58 | 8.03 |
> | 1B | DeBLAS | 62.5 | 16.44 | 19.04 | 3.57 | 13.95 | 11.81 | 8.46 | 8.10 |
>
> In the revised version, we have revised the broader impact discussion to avoid this unsupported claim. We have instead clarified that our method does not explicitly manipulate semantic content, while acknowledging that evaluating the impact of length-based scheduling on representation bias remains an important direction for future work.

---

> > ### Author Response · Authors · 2026-06-01
> >
> > ### On the weakness about hyperparameter selection
> >
> > > No discussion or guidelines on selecting hyperparameters for a practical application.
> >
> > We thank the reviewer for raising this important practical concern regarding hyperparameter selection. Several of these hyperparameters have already been investigated in the current submission, while others were not discussed in sufficient detail. In the revised version of our manuscript, we have expanded and provided a systematic set of selection guidelines for all the DeBLAS hyperparameters in Appendix A.2.2.
> >
> > - Number of dense-batching iterations ($T_d$): Intuitively, if the $T_d$ is too small, the model cannot fully benefit from the maximized token utilization of dense batches; if it is too large, the accumulated length-wise bias requires a longer correction stage afterward. Based on this intuition, we propose a practical heuristic: the transition to the Balanced Batching stage should occur when the training loss of the Dense Batching stage begins to plateau. Specifically, in our experiments, we record the training loss every 5% of the total training iterations and define the onset of the plateau as a loss change smaller than 0.1 compared to the previous checkpoint. Following this criterion, we set $T_d$ to approximately 40% of the iterations required for regular training to reach the target validation perplexity in the main experiments.
> >
> > - Number of length bins ($K$): Figure 5(c) compares multiple length bin partition strategies (Equal-3, Equal-5, Exp-4). The results show that DeBLAS is robust across different binning schemes, suggesting that the method is not sensitive to the exact choice of K. Empirically, we find that using 3-5 bins provides a good trade-off between calibration granularity and implementation simplicity.
> >
> > - Uniform length of dense batch ($L_d$): Figure 5(d) studies different values of $L_d$. The results indicate that choosing $L_d$to be at least half of the model context length yields consistently strong performance, whereas excessively short $L_d$ significantly degrades convergence due to insufficient modeling of long-range dependencies. In other words, given the model context length $L$, a choice of $L_d$ from a range of $[L/2, L]$ would be recommended.
> >
> > - Base batch size $B$: The base batch size is not a DeBLAS-specific hyperparameter, but a standard pretraining configuration. The token batch size $N_B$ satisfies $N_B=BL$, where $B$ is the base batch size and $L$ is the model context length. In all experiments, both the token batch size and context length are configured following prior works [1,2]. Therefore, no additional tuning of the base batch size is introduced by our method.
> >
> > - Calibration set size ($N_C$): Conceptually, the calibration set $D_C$ only needs to approximate the global length distribution of the pretraining corpus. We have additionally conducted sensitivity experiments with different calibration set sizes while keeping all other settings fixed in Table 9, Appendix A.2.2 of the revised manuscript (also shown below). We empirically find that a relatively small subset (e.g., around 1K samples) already provides good acceleration performance.
> >
> > **Table. Sensitivity analysis of $N_C$ with LLaMA 130M pretrained on C4. The number of required iterations (in thousands) to reach the target PPL of 25.4 is reported.**
> > |$N_C$ | 200 | 1000 | 2000 | 5000 |
> > |---|---:|---:|---:|---:|
> > | Iterations | 16 | 13 | 13 | 13 |
> >
> > - Calibration frequency ($T_C$): Conceptually, very frequent recalibration increases evaluation overhead without noticeable performance gains, whereas overly infrequent updates slow the adaptation to length-wise bias. Empirically, we have performed additional ablations on the calibration frequency $T_C$ in Table 10, Appendix A.2.2 of the revised manuscript (also shown below). The results show that DeBLAS is similarly robust to the calibration frequency. Updating every 1K iterations already achieves a favorable balance between convergence performance and computational efficiency.
> >
> > **Table. Sensitivity analysis of $T_C$ with LLaMA 130M pretrained on C4. The number of required iterations (in thousands) to reach the target PPL of 25.4 is reported.**
> > |$T_C$ | 500 | 1000 | 2000 | 4000 |
> > |---|---:|---:|---:|---:|
> > | Iterations | 13 | 13 | 14 | 16 |
> >
> > [1] Zhao, Jiawei, et al. "GaLore: Memory-Efficient LLM Training by Gradient Low-Rank Projection." International Conference on Machine Learning. PMLR, 2024.
> >
> > [2] Han, Andi, et al. "SLTrain: a sparse plus low rank approach for parameter and memory efficient pretraining." Advances in Neural Information Processing Systems 37 (2024): 118267-118295.

---

> > > ### Author Response · Authors · 2026-06-01
> > >
> > > ### Requested changes
> > >
> > > > The end of Section 2 introduces token utilization rate. It states, "We empirically demonstrate that packing can actually reduce training efficiency compared to padding under a fixed token batch size (refer to Appendix A.3.1)." None of the tables or figures in the manuscript, including appendixes, ever demonstrate that. Neither result seems to use a token utilization rate. Why does this metric need to be introduced? Could you show that "packing can actually reduce training efficiency compared to padding"?
> > >
> > > Thank you for the detailed feedback. Our intention about this statement is to claim that packing may require more optimization iterations for the model to reach the same validation perplexity compared to padding under a fixed token batch size, as observed in Figure 7 of Appendix A.3.1. Our intention in introducing TUR was to provide an intuitive analytical perspective for understanding how sequence length and batching strategies affect the proportion of semantically meaningful token interactions within a fixed token budget. It is used as a motivating concept for the Dense Batching stage, as explained in Section 3.1 of the revised manuscript, rather than as a directly optimized training objective or a standalone performance metric. In causal language modeling, Dense Batching achieves the highest possible TUR. It eliminates padding tokens while avoiding cross-document concatenation and the resulting interactions between unrelated documents. Therefore, all attention computations are devoted to semantically coherent token interactions. This observation motivates our design choice of using dense batches in the first stage to maximize the effective utilization of the available training budget.
> > >
> > > To improve clarity, we have modified the original statement as “Empirically, in Appendix A.3.1, we observe that packing requires more optimization iterations than padding for the model to reach the same validation perplexity under a fixed token batch size.”
> > >
> > > > Generally, there are several claims in the text that are not supported by any results. I encourage authors to go through the text and make sure that if it is written as "We prove...", "We demonstrate...", or "our work does...", there are results that prove, demonstrate, and do.
> > >
> > > Thank you for the constructive comment. We have carefully checked the manuscript and revised some parts to ensure that the potential claims are aligned with the actual scope of the presented analyses and experiments, and are described in a proper manner. The detailed revisions have been explained above in our responses to the corresponding unsupported claims. (also refer to Abstract, Section 2, Broader Impact Statement, and Appendix A.3.1 of the revised manuscript)
> > >
> > > > Fig. 3c is information dense and has an unfortunate colour scheme. From the description (including the appendix), I cannot understand what it shows. Consider recolouring one of the light greens. Besides that, it is unclear what "short-grad" or "short-loss" means. For example, "short-loss" never appears elsewhere in the text. Also, please specify that the details of this figure are elaborated in Appendix A.3.2.
> > >
> > > Thank the reviewer for the valuable suggestion. “Short/Middle/Long” in the figure legend refers to different models trained exclusively on sequences from the corresponding length bins. Concretely:
> > > - “Short” denotes the model trained only on the [0,127] length bin,
> > > - “Middle” denotes the model trained only on the [128,255] length bin,
> > > - “Long” denotes the model trained only on the [256] length bin.
> > >
> > > The curves labeled “Short-Loss”, “Middle-Loss”, and “Long-Loss” represent the evaluation losses of these three models on different length bins. Similarly, the bars labeled “Short-Grad”, “Middle-Grad”, and “Long-Grad” represent the corresponding gradient norms for these three models calculated on different length bins.
> > >
> > > The purpose of Figure 3(c) is twofold:
> > > 1. **Demonstrating length-wise generalization bias induced by single-length training.**
> > > The loss curves show that each model achieves the lowest loss on the length range it was trained on, while suffering noticeably higher loss on out-of-bin sequence lengths. This observation motivates the Balanced Batching stage in DeBLAS.
> > >
> > > 2. **Motivating the use of loss as a surrogate for gradient norm in adaptive sampling.**
> > > The gradient norm bars and loss curves exhibit a consistent positive correlation across length bins and models: bins with larger losses also tend to produce larger gradient norms. This empirically supports our design choice of using training loss to adjust sampling probabilities during the Balanced Batching stage.
> > >
> > > In the revised version, we have redesigned the figure with a more distinguishable color scheme and clearer explanations in the caption and the main text. We have also explicitly stated both in the caption and the main text that the experimental setup and detailed interpretation of Figure 3(c) are provided in Appendix A.3.2.

---

> > > > ### Author Response · Authors · 2026-06-01
> > > >
> > > > > The experiments are reporting a decrease in the number of iterations. Table 4 also reports a wall-clock training time. These are great results. However, for the practical engineering applications, the memory-related metrics are also representative. Could you report and compare to the random sampling: (1) GPU utilization for both stages; (2) Peak GPU memory utilization for both stages.
> > > >
> > > > We thank the reviewer for this valuable suggestion. Conceptually, DeBLAS does not increase the overall token budget per iteration. Instead, it improves training efficiency by scheduling sequence length and enhancing token utilization. Therefore, its memory footprint should remain largely comparable against the Random baseline.
> > > >
> > > > To further validate this, we conduct profiling experiments on a single NVIDIA A100 GPU using LLaMA 130M model and C4 dataset, following the same setup as the main experiments of Table 1. We separately measure GPU utilization and peak GPU memory usage for the Dense Batching stage and the Balanced Batching stage in Table 5 of the revised manuscript (also shown below). The details of the profiling experiment are provided in the Appendix A.3.7 of the revised manuscript.
> > > >
> > > > **Table. Memory efficiency analysis of DeBLAS.**
> > > > | Method | GPU Utilization (%) | Peak GPU Memory (GB) |
> > > > | ------ | -------------------:| --------------------:|
> > > > | Random | 90.26 | 46.21 |
> > > > | DeBLAS-Dense | 90.32 | 46.10 |
> > > > | DeBLAS-Balanced | 89.35 | 46.88 |
> > > >
> > > > The experiment results show that both stages achieve comparable GPU utilization and peak GPU memory usage against Random, indicating that the online calibration and dynamic sampling of the Balanced Batching stage introduce negligible runtime overhead and memory consumption. We have included these profiling results and discussions in a dedicated subsection of Section 4.3 of the revised manuscript.
> > > >
> > > > > The paper presents several additional hyperparameters: base batch size, number of dense batching iterations, number of bins, uniform length of dense batch, and calibration set size. There is no instruction on how to choose them if the methodology is applied for pretraining. Could you discuss a set general guidelines on how to select those parameters?
> > > >
> > > > Thank you for the valuable suggestion. Detailed discussions regarding the selection principles and practical recommendations for the additional hyperparameters have been provided above in our response to the comment concerning the weakness about hyperparameter selection. In the revised manuscript, we have incorporated these recommendations and discussions in Appendix A.2.2.
> > > >
> > > > > Table 4. Please specify that a number of iterations are reported in thousands. Otherwise, the math does not make sense.
> > > >
> > > > Thank the reviewer for catching this ambiguity. We have revised the table caption and column header to clarify this in the revision.
> > > >
> > > > > The Broader Impact Statement is generally for any direct ethical implications of your work. Please do not treat it as a second conclusion.
> > > >
> > > > We thank the reviewer for this helpful suggestion. In the revised manuscript, we have substantially revised the Broader Impact Statement to better align with the NeurIPS/TMLR guidelines by focusing on the potential ethical implications of length-based scheduling strategies.

---

> > > > > ### Comment · Reviewer_VFTT · 2026-06-27
> > > > > **Response to Authors**
> > > > >
> > > > > Thank you for your hard work. In the updated version of the manuscript all the presented claims are sufficiently supported.
> > > > >
> > > > > I especially thank you for putting together a comprehensive hyperparameter guidelines. This makes your work significantly more approachable for engineering people, who would look to apply it on practice.
> > > > >
> > > > >
> > > > > ## Comments on changes
> > > > >
> > > > > ### TUR (eq 1)
> > > > > > Our intention in introducing TUR was to provide an intuitive analytical perspective for understanding...
> > > > > > It [TUR] is used as a motivating concept for the Dense Batching stage...
> > > > >
> > > > > In other words, TUR is not so valuable for the rest of the paper. Especially since you have a well-written motivation in section 3, that includes experiments and theory.
> > > > >
> > > > > If you want to keep it, please move eq 1 and it description to section 3, specifically subsection "Maximization of the TUR via dense batches". That would improve readability.
> > > > >
> > > > > ### Memory utilization
> > > > >
> > > > > Thank you for showing this result. Lack of memory overhead is a great selling point for the engineering applications.
> > > > >
> > > > > ### The Broader Impact Statement
> > > > >
> > > > > The current statement is describing limitations and future work direction, rather than the broader impact. However, it is not a big issue, since your work does not bring ethical concerns. So you may leave it as is.

---

> > > > > > ### Author Response · Authors · 2026-06-28
> > > > > >
> > > > > > We sincerely thank the reviewer for the careful follow-up evaluation and the encouraging comments! We also appreciate the suggestion about TUR and will incorporate the suggested changes in the revised manuscript.

---

### Review · Reviewer_gEo9 · 2026-05-18

**Summary Of Contributions:**

This paper proposes Dense-Balanced Sequence Scheduling (DeBLAS), a two-stage sequence-length-based data scheduling method for accelerating LLM pretraining. The first stage trains on dense uniform-length batches to improve token utilization and accelerate early representation learning. The second stage switches to variable-length padded batches, where sampling probabilities over length bins are periodically adjusted using held-out losses to reduce the length-wise bias introduced by dense batching.

The main strengths are that the method is simple, practical, does not require a reference model or expensive semantic/influence estimation, and is evaluated across multiple datasets, model sizes, and ablations. The paper provides useful evidence that dense batching improves early training efficiency but can cause length-wise generalization bias, and that the balanced stage helps correct this issue.

The main weaknesses are that the headline acceleration claims are still presented mostly as reductions in training iterations, while the useful wall-clock measurements are concentrated in selected comparisons rather than consistently aligned with the main headline settings; the novelty is moderate relative to prior length-curriculum, Dataset Decomposition/VSL-style training, token-utilization-aware batching, and loss-based sampling; and the theoretical analysis relies on strong assumptions and is only loosely connected to practical transformer pretraining.

Overall, I find the paper practically useful and relevant to efficient LLM pretraining, though several claims would benefit from stronger compute-based evidence and clearer separation between calibration and evaluation.

**Additional Comments:**

I enjoyed reading the paper. The method is simple, practical, and motivated by a real issue in LLM pretraining pipelines. I especially appreciated the ablations showing why both stages are needed: dense batching improves early token utilization, while balanced batching helps correct the resulting length-wise bias.

My main concern is that the paper currently makes a strong acceleration argument whose headline evidence is mostly iteration-count based, even though selected runtime comparisons and overhead analyses are included. I think the work would be substantially stronger if the authors reported end-to-end wall-clock or compute-normalized savings more consistently across the main settings. With that clarification, I would view the paper as a useful contribution to efficient LLM pretraining.

**Audience:**

Yes

**Audience Explanation:**

Yes. Efficient LLM pretraining is an important topic for a substantial part of the TMLR audience, especially researchers working on data selection, curriculum learning, batching strategies, training efficiency, and scalable language model training. The proposed method is simple and practically motivated: sequence length is easy to measure, and the method does not require a reference model, gradient matching, influence estimation, or domain annotations.

The paper is also interesting because it highlights a concrete tension in pretraining data construction: dense batches improve token utilization and early efficiency, but can induce length-wise generalization bias if used alone. This observation is useful even beyond the specific DeBLAS algorithm, since it may inform future work on packing, padding, context-length curricula, and data scheduling.

The findings are likely to be of interest to both ML researchers and practitioners building large-scale training pipelines, provided that the broad efficiency claims are interpreted primarily as iteration reductions except for the specific comparisons where wall-clock evidence is reported.

**Broader Impact Concerns:**

I do not see major broader impact concerns beyond those already associated with efficient LLM pretraining. The method may reduce the computation required to reach a given pretraining perplexity, which could lower energy usage and cost. At the same time, more efficient training methods can also make it easier to train larger or more numerous models, with the usual concerns around misuse, environmental cost, and propagation of biases from web-scale corpora.

The paper includes a Broader Impact Statement, and I think it is generally adequate. One possible addition would be to mention that length-based scheduling may change the effective distribution of training data if sequence length correlates with domain, source, or style. This could potentially affect the representation of different data types, although the paper does include some domain-level evaluation.

**Claims And Evidence:**

Yes

**Claims Explanation:**

The main empirical claim, that DeBLAS can reach comparable validation perplexity with fewer optimization steps than standard random sampling, is supported by a broad set of experiments on C4 and SlimPajama, across LLaMA-style models from 60M to 1B parameters, with an additional 7B-scale experiment. The ablations are also useful: they show that dense batching alone introduces length-wise bias, balanced batching alone is less effective, and the two-stage method performs better than either component in isolation. The comparisons with random sampling, several online data selection baselines, Dataset Decomposition, and reference-model-based methods further strengthen the empirical case.

However, I think the evidence is less complete for the stronger framing that DeBLAS reduces the overall computational burden of pretraining across settings. The paper includes useful runtime measurements for selected comparisons, including reference-model-based baselines, Dataset Decomposition variants, and calibration overhead, but these measurements are not consistently aligned with the main headline settings in Table 1 across datasets and model sizes, nor with the larger-scale 7B and long-context experiments. Since DeBLAS changes sequence lengths, padding behavior, token utilization, and calibration overhead, iteration count alone is not always equivalent to wall-clock time, FLOPs, or total useful tokens processed.

I also think the theory should be interpreted cautiously. The assumptions, such as token-level independence and positive correlation between loss and gradient norm, are strong and do not fully capture transformer pretraining dynamics. The theoretical section is useful as intuition, but it is not the main source of evidence for the paper’s claims.

Thus, I would answer “yes” for the core empirical claim about fewer optimization steps to comparable perplexity, while recommending that the authors qualify or strengthen the broader compute-efficiency claims with more systematic end-to-end evidence.

**Requested Changes:**

### Critical to my recommendation

1. **Extend the existing runtime and overhead analyses more systematically.**
   The paper includes useful runtime evidence in selected settings, including comparisons with reference-model-based methods, Dataset Decomposition variants, and calibration overhead. However, the main headline results in Table 1, as well as the 7B and long-context experiments, are still primarily reported as reductions in optimization iterations. Since DeBLAS changes sequence-length composition, padding behavior, token utilization, and calibration overhead, I recommend reporting end-to-end wall-clock time, FLOPs, or token-level compute for the main headline settings, or clearly qualifying those claims as iteration-efficiency results. If the demonstrated benefit in a given setting is fewer optimization steps rather than directly measured lower wall-clock time, the paper should state this clearly and tie phrases such as “reduce computational burden” to settings where runtime evidence is available.

2. **Clarify the separation of calibration, training, and evaluation data.**
   The Balanced Batching stage adjusts length-bin sampling probabilities using evaluation losses computed on a calibration set $D_C$ (Page 5, Section 3.2), which the paper describes as "a small subset of training samples" (Page 5). However, the paper does not explicitly state whether (a) this calibration subset is excluded from the training batches used for optimization, or (b) whether $D_C$ is disjoint from the validation set used to define target perplexity in Table 1 and Table 2. Please clarify these data splits to ensure there is no overlap between data used for scheduling, validation, and downstream evaluation.

3. **Qualify the theoretical claims.**
   The theoretical analysis relies on strong assumptions, including token-level independence and positive correlation between loss and gradient norm. These assumptions are useful for intuition but do not fully model transformer pretraining. I recommend softening claims that the theory “proves” the practical mechanism, and instead presenting it as an idealized analysis that supports the intuition behind the two-stage design.

4. **Moderate claims about downstream improvements and clarify novelty relative to prior work.**
   The downstream benchmark results are generally comparable or slightly better than random sampling, but many improvements are modest. The paper should primarily position DeBLAS as an efficiency method rather than substantially improving downstream task accuracy. Additionally, it would be helpful to clarify the technical distinction between DeBLAS and prior length-curriculum methods (e.g., whether the transition is fixed-schedule or data-driven), and whether DeBLAS's advantage persists when compared against modern length-aware packing implementations rather than simple padding baselines.

### Would strengthen the work

5. **Add sensitivity analysis for calibration-specific hyperparameters.**
   The paper already studies transition timing, length-bin partitioning, and dense-stage length. It would further strengthen the method to analyze calibration set size $N_c$, calibration frequency $T_c$, and the stability of learned length-bin probabilities across seeds, datasets, and model sizes.

---

> ### Author Response · Authors · 2026-06-01
>
> ### Requested changes
> > 1. Extend the existing runtime and overhead analyses more systematically.
>
> We thank the reviewer for the insightful feedback. In the revised manuscript, we have substantially extended the efficiency analysis by additionally reporting the end-to-end GPU hours required to reach the target validation perplexity for the main settings in Table 1 of the manuscript. All experiments are conducted on NVIDIA A100 GPUs. The updated results are summarized below.
>
> **Table. End-to-end GPU hours required for different methods in LLM pretraining to reach the given validation perplexity (PPL).**
> |            |     | C4   |      |     |    |     | SlimPajama   |      |     |
> | ---------- | --- | ---- | ---- | --- | --- | --- | ---- | ---- | --- |
> | **Model Size** | **60M** | **130M** | **350M** | **1B**  | **Model Size** | **60M** | **130M** | **350M** | **1B**  |
> | **Target PPL** | **30.4** | **25.4** | **18.8** | **16.9** | **Target PPL** | **26.5** | **21.7** | **17.6** | **15.5** |
> | **Method** | | | | | **Method**|
> | Random | 1.56 (-) | 6.22 (-) | 65.12 (-) | 569.34 (-) | Random | 3.19 (-) | 12.89 (-) | 65.98 (-) | 571.96 (-) |
> | Longest | 1.56 (1.00x) | 4.35 (1.43x) | 71.63 (0.91x) | 540.87 (1.05x) | Longest | 3.19 (1.00x) | 14.50 (0.89x) | 71.48 (0.92x) | >R |
> | Max Loss | 3.99 (0.39x) | 16.34 (0.38x) | 171.00 (0.38x) | >R | Max Loss | 8.97 (0.36x) | 35.06 (0.37x) | 175.13 (0.38x) | >R |
> | Max Grad | 3.60 (0.43x) | 10.46 (0.59x) | 177.88 (0.37x) | >R | Max Grad | 4.98 (0.64x) | 26.85 (0.48x) | 91.28 (0.72x) | >R |
> | FM | 3.38 (0.46x) | 11.34 (0.55x) | 109.89 (0.59x) | >R | FM | 5.32 (0.60x) | 32.61 (0.40x) | 106.90 (0.62x) | >R |
> | GREATS | 3.88 (0.40x) | 11.22 (0.55x) | 143.82 (0.45x) | >R | GREATS | 5.58 (0.57x) | 21.89 (0.59x) | 93.48 (0.71x) | 907.87 (0.63x) |
> | **DeBLAS** | **1.24 (1.26x)** | **4.02 (1.54x)** | **43.25 (1.51x)** | **341.04 (1.67x)** | **DeBLAS** | **2.20 (1.45x)** | **9.02 (1.43x)** | **42.89 (1.54x)** | **357.47 (1.60x)** |
>
> The newly added runtime results show that the iteration reduction achieved by DeBLAS translates almost directly into end-to-end GPU-hour reduction across different model scales and datasets. This is because DeBLAS introduces nearly no additional per-iteration computational overhead. Specifically:
> 1. DeBLAS keeps the token batch size fixed throughout training, although it changes sequence-length composition and padding behavior.
> 2. The method does not require additional candidate scoring passes, reference-model evaluations, or extra gradient computations, which are the major sources of overhead in methods such as Max Loss, Max Grad, FM, and GREATS.
> 3. As empirically shown in Appendix A.3.3 of the revised manuscript, the periodic calibration process in the Balanced Batching stage incurs only negligible runtime overhead compared to the overall pretraining process.
>
> Therefore, for DeBLAS, reducing the number of optimization iterations is a reliable proxy for reducing actual pretraining time cost, which validates the use of iteration reduction as a measure of pretraining acceleration in the original manuscript. In contrast, the newly added GPU-hour results also clarify an important distinction for prior online selection baselines: several methods occasionally reduce optimization iterations but still require substantially larger end-to-end runtime due to their expensive candidate scoring and selection procedures.
>
> Furthermore, we have also modified Figure 4(a) (the 7B model scaling experiment) with GPU-hour comparisons. For the long-context experiment in Figure 4(c), we additionally report the corresponding end-to-end GPU hours in Appendix A.3.5, where Random and DeBLAS require 275.39 and 169.12 GPU hours, respectively. Both experiment results demonstrate that DeBLAS achieves great runtime acceleration performance.
>
> > 2. Clarify the separation of calibration, training, and evaluation data.
>
> Thank the reviewer for the valuable suggestion. To clarify, the calibration set $D_C$ used in the Balanced Batching stage is completely disjoint from both the training batches, the validation set, and all downstream evaluation data. In the revised version, we have explicitly stated in Section 3.2 that there is no overlap between data used for periodic calibration, model optimization, validation, or downstream evaluation.
>
> > 3. Qualify the theoretical claims.
>
> Thank the reviewer for this insightful feedback. We agree that the theoretical analysis relies on several simplifying assumptions that may not fully apply to practical transformer pretraining. In the revised version, we have softened claims such as “prove” or “theoretically demonstrate” and instead present the analysis more explicitly as an idealized framework that provides intuition and supporting evidence for the empirical design choices.

---

> > ### Author Response · Authors · 2026-06-01
> >
> > > 4. Moderate claims about downstream improvements and clarify novelty relative to prior work.
> >
> > We would like to thank the reviewer for this valuable comment.
> >
> > **On downstream benchmark results**
> >
> > We would like to clarify that our intention was not to claim substantial downstream accuracy improvements over random sampling. Instead, the downstream benchmark experiments are intended to demonstrate that DeBLAS can achieve comparable downstream performance using substantially fewer pretraining iterations. We have made this positioning more explicit in Section 4.2 of the revised manuscript.
> >
> > **On distinction from prior length-curriculum methods**
> >
> > We appreciate the opportunity to explain this point more clearly. Existing length-curriculum approaches, such as GrowLength and Dataset Decomposition (DD), rely on predefined length schedules. GrowLength progressively increases the training sequence length according to a fixed curriculum, while DD samples from predefined length buckets (i.e., $2^i$) following a fixed curriculum. In contrast, DeBLAS employs a two-stage strategy in which the second stage dynamically adjusts length-bin sampling probabilities according to online loss statistics collected during training. The second stage is therefore data-driven and adaptive.
> >
> > Furthermore, as discussed in Section 3.1, Section 4.2, and Appendix A.3.6, both GrowLength and DD overlook the length-wise bias induced by forcing training batches to remain within a narrow length range. In contrast, DeBLAS exposes the model to all sequence lengths during training through the Balanced Batching stage, thereby mitigating the bias introduced by the Dense Batching stage. To clarify the distinction from prior length-curriculum methods, we have added a dedicated subsection “Length-based curriculum” in Appendix B of the revised manuscript.
> >
> > **On comparison against modern length-aware packing implementations**
> >
> > As discussed in Section 2, although packing eliminates the need for padding tokens, it may also reduce token utilization efficiency when complete documents are split into multiple segments and introduce cross-document contamination when segments of different documents are packed into the same sequence. Under a fixed token batch size, this fragmentation can decrease the amount of semantically coherent context available during training. Our empirical comparison in Appendix A.3.1 shows that packing does not improve training efficiency relative to padding in this setting. In contrast, DeBLAS is designed to improve training efficiency from the perspective of both token utilization and model generalization. The Dense Batching stage improves TUR by reducing unnecessary padding within batches, while the Balanced Batching stage further improves optimization efficiency and mitigates length-wise bias through adaptive loss-aware scheduling across all sequence lengths.
> >
> > In the revised version, we have further strengthened the discussion of the relationship between DeBLAS and modern packing-based training strategies in Appendix A.3.1 to better clarify the novelty and scope of the proposed method.

---

> ### Author Response · Authors · 2026-06-01
>
> > 5. Add sensitivity analysis for calibration-specific hyperparameters.
>
> Thank the reviewer for the valuable suggestion.
>
> - Calibration set size ($N_C$): Conceptually, the calibration set $D_C$ only needs to approximate the global length distribution of the pretraining corpus. We have conducted sensitivity experiments with different calibration set sizes ($N_C$) while keeping all other settings fixed in Table 9, Appendix A.2.2 of the revised manuscript (also shown below). We empirically find that DeBLAS is not particularly sensitive to the exact calibration set size once a moderate number of samples is used. A relatively small subset (e.g., around 1K samples) already provides good performance while introducing negligible computational overhead.
>
> **Table. Sensitivity analysis of $N_C$ with LLaMA 130M pretrained on C4. The number of required iterations (in thousands) to reach the target PPL of 25.4 are reported.**
> |$N_C$ | 200 | 1000 | 2000 | 5000 |
> |---|---:|---:|---:|---:|
> | Iterations | 16 | 13 | 13 | 13 |
>
> - Calibration frequency ($T_C$): We have performed additional ablations on the calibration frequency ($T_C$) in Table 10, Appendix A.2.2 of the revised manuscript (also shown below). The results show that DeBLAS is similarly robust to the calibration frequency $T_C$. In practice, updating every 1K iterations already provides a favorable trade-off between convergence performance and computational efficiency.
>
> **Table. Sensitivity analysis of $T_C$ with LLaMA 130M pretrained on C4. The number of required iterations (in thousands) to reach the target PPL of 25.4 are reported.**
> |$T_C$ | 500 | 1000 | 2000 | 4000 |
> |---|---:|---:|---:|---:|
> | Iterations | 13 | 13 | 14 | 16 |
>
>
> **On stability of learned length-bin probabilities**
>
> We have conducted additional experiments to analyze the stability of the learned length-bin sampling probabilities across random seeds, datasets, and model scales. Specifically, we evaluate DeBLAS on both C4 and SlimPajama using LLaMA models of sizes 60M, 130M, and 350M, with three different random seeds for each configuration. Due to the substantial computational cost of conducting multiple repeated runs at the 1B scale, we currently limit this stability analysis to models up to 350M parameters.
>
> During the Balanced Batching stage, we record the sampling probability of every length bin at each calibration step. For each configuration, we compute the standard deviation of the learned sampling probabilities across random seeds and then average the resulting values over all length bins and calibration steps. The results of Table 6 of the revised manuscript (also shown below) show consistently low variability across all evaluated considerations.
>
> **Table. Average standard deviation of learned length-bin sampling probabilities across random seeds, computed over all length bins and calibration steps.**
> | Dataset | 60M | 130M | 350M |
> | --------- | ---:| ----:| ----:|
> | C4        |  0.013 |  0.011 | 0.012 |
> | SlimPajama|  0.016 |  0.012 | 0.014  |
>
> We have added a new subsection, “Stability of Learned Length-Bin Sampling Probabilities”, in Section 4.3 in the revised manuscript to present this analysis more clearly.
>
> > The paper includes a Broader Impact Statement, and I think it is generally adequate. One possible addition would be to mention that length-based scheduling may change the effective distribution of training data if sequence length correlates with domain, source, or style. This could potentially affect the representation of different data types, although the paper does include some domain-level evaluation.
>
> We thank the reviewer for this valuable suggestion regarding the broader impact discussion. We agree that length-based scheduling could potentially alter the effective distribution of training data when sequence length correlates with factors such as domain, source, or writing style. We have explicitly acknowledged this potential impact in the revised broader impact statement.

---

### Review · Reviewer_rUFL · 2026-06-13

**Summary Of Contributions:**

The major conribution is Token Utilization Rate (TUR) as a lens for pretraining efficiency. The framing padding waste as a quantifiable metric is a clean and useful conceptual contribution, even if the solution (dense batching) is fairly straightforward once you have this framing.

This paper identifies and quantifies that models trained on narrow length ranges generalize poorly to other lengths (Figure 3c) is a useful finding for the community, regardless of whether DeBLAS is the right solution.

The two-stage curriculum idea also makes sense, which combins dense batching with adaptive loss-based length rebalancing is a reasonable and practical design, with solid ablation support.

**Audience:**

Yes

**Audience Explanation:**

The paper focus on the important topic is llm training efficiency

**Claims And Evidence:**

Yes

**Claims Explanation:**

The results are mostly reliable

**Requested Changes:**

1. Add a fair packing baseline
This is the most critical missing experiment. The entire motivation of the paper rests on padding being the default, but modern pretraining uses packing with attention masks to prevent cross-document contamination. Without showing DeBLAS outperforms this, the core premise is undermined. The authors need to either add this comparison or explicitly scope the paper to padding-based pretraining and justify why that's a meaningful setting.

2. Longer context length experiments
The main experiments use context length 256-512, which is far from modern pretraining practice. The method's effectiveness at 4K, 8K, or longer context windows needs to be demonstrated, especially given the concern that Stage 1 training on L/2 sequences could hurt long-range dependency learning. Algorithm 2 is proposed for this setting but is very thinly validated.

3. Disentangle the two stages' contributions more carefully
The current ablation compares DeBLAS vs only-Stage2 vs only-Stage1, but doesn't isolate whether Stage 1's benefit comes from the dense batching itself or from serving as a warm-up for Stage 2. A cleaner ablation would be: random sampling for 40% of iterations then Stage 2, vs DeBLAS. This would show whether Stage 1 specifically is necessary or just any warm-up phase.

---

> ### Author Response · Authors · 2026-06-24
>
> ### Requested Changes
> > Add a fair packing baseline This is the most critical missing experiment. The entire motivation of the paper rests on padding being the default, but modern pretraining uses packing with attention masks to prevent cross-document contamination. Without showing DeBLAS outperforms this, the core premise is undermined. The authors need to either add this comparison or explicitly scope the paper to padding-based pretraining and justify why that's a meaningful setting.
>
> We thank the reviewer for this valuable suggestion. Our work focuses on **data scheduling** to accelerate language model pretraining. Padding and packing represent different **data preprocessing strategies** rather than different **scheduling policies**. The motivation of DeBLAS is that sequence length constitutes an informative scheduling signal when complete sequences remain explicitly observable. Packing concatenates multiple documents and splits them into fixed-length segments, which obscures individual sequence boundaries and makes sequence-level length information unavailable to the scheduler. In the current manuscript, we intentionally restrict our main experiment to the padding-based setting to isolate the acceleration benefits brought by sequence-length scheduling.
>
> Furthermore, following the reviewer's suggestion, we **have additionally implemented a modern packing baseline** that employs attention masks to prevent cross-document attention. We have conducted experiments on both C4 and SlimPajama using LLaMA 60M and 130M models to directly compare it with DeBLAS. The results are shown in **Table 11, Appendix A.3.1**, of the revised manuscript (also shown below).
>
> **Table. Training Iterations (x1000) ($\downarrow$) required for Packing and DeBLAS to reach the given validation perplexity (PPL). Values of relative speedup against Packing ($T_{\rm Packing}/T_{m}$, $\uparrow$) are reported in brackets.**
> | Dataset | C4 | C4 | SlimPajama | SlimPajama |
> |--------|----:|----:|-----------:|-----------:|
> | Model size | 60M | 130M | 60M | 130M |
> | Target PPL | 30.4 | 25.4 | 26.5 | 21.7 |
> | Packing | 11 (-) | 22 (-) | 21 (-) | 41 (-) |
> | **DeBLAS** | **8 (1.38×)** | **13 (1.69×)** | **14 (1.50×)** | **28 (1.46×)** |
>
> The results show that DeBLAS consistently achieves better training efficiency than packing across all datasets and model sizes. We conjecture that the document fragmentation introduced by packing is an important factor for this performance gap. Although attention masks successfully prevent cross-document contamination, documents are frequently split across multiple packed sequences. Consequently, many tokens lose access to their original long-range context, which may interrupt semantic continuity and weaken long-context dependency modeling. This observation is also consistent with our token utilization rate (TUR) formulation in Eq. (1). After document splitting, the number of semantically meaningful preceding tokens that each token can attend to decreases substantially, leading to lower TUR. Therefore, while packing eliminates padding tokens, it may still reduce optimization efficiency compared to DeBLAS.
>
> We have incorporated these results and discussions in Appendix A.3.1 of the revised manuscript.

---

> > ### Author Response · Authors · 2026-06-24
> >
> > > Longer context length experiments The main experiments use context length 256-512, which is far from modern pretraining practice. The method's effectiveness at 4K, 8K, or longer context windows needs to be demonstrated, especially given the concern that Stage 1 training on L/2 sequences could hurt long-range dependency learning. Algorithm 2 is proposed for this setting but is very thinly validated.
> >
> > We thank the reviewer for this constructive feedback. The original submission already includes a long-context experiment using a context length of 2048 in Section 4.2. Following the reviewer's suggestion, we have further conducted an additional experiment using a context length of 4096 in **Table 13, Appendix A.3.4**, of the revised manuscript (also shown below).
> >
> > **Table. Evaluation on a context length of 4096. Training Iterations (x1000) (↓) to reach the given validation perplexity (PPL) and values of relative speedup against Random are reported.**
> > | Target PPL | Method | Iterations | Speedup |
> > | ---------- | ------:| ----------:| -------:|
> > | 15.2       | Random |         50 |       - |
> > | 15.2       | DeBLAS |         31 |   1.61x |
> >
> > The additional results show that DeBLAS continues to substantially reduce the number of optimization iterations required to reach the target validation perplexity, demonstrating that the proposed scheduling strategy remains effective at long context windows.
> >
> > Regarding the concern that the Dense Batching Stage may hurt long-range dependency learning, we would like to clarify that the Dense Batching stage of Algorithm 2 does not train exclusively on a fixed length of $L/2$. The model is first trained on dense batches composed of shorter sequences and subsequently exposed to dense batches  composed of increasingly longer sequences, eventually reaching the full context length before entering the Balanced Batching stage. Therefore, long-context sequences are already introduced during this stage. Furthermore, the subsequent Balanced Batching stage restores training using the full context window. Since the sampling probabilities are periodically adjusted according to the evaluation losses of different length bins, underperforming long-sequence bins automatically receive higher sampling probabilities. Consequently, any insufficient long-context modeling after the Dense Batching stage is progressively corrected during the Balanced Batching stage.
> >
> > We have added these results and discussions in Appendix A.3.4 of the revised manuscript.
> >
> > > Disentangle the two stages' contributions more carefully The current ablation compares DeBLAS vs only-Stage2 vs only-Stage1, but doesn't isolate whether Stage 1's benefit comes from the dense batching itself or from serving as a warm-up for Stage 2. A cleaner ablation would be: random sampling for 40% of iterations then Stage 2, vs DeBLAS. This would show whether Stage 1 specifically is necessary or just any warm-up phase.
> >
> > We thank the reviewer for this insightful suggestion. Following the reviewer's suggestion, we have conducted a stronger control experiment in which the model is trained with standard random sampling during the first 40% of training iterations and then switches to the Balanced Batching stage used by DeBLAS and added the result in **Figure 5(b), Section 4.3**, of the revised manuscript (denoted as Random→Balanced).
> >
> > The results show that this Random→Balanced variant significantly underperforms DeBLAS, which demonstrates that the Dense Batching stage itself plays an essential role. This result is consistent with our empirical exploration in Section 3.1. Dense batches maximize token utilization and reduce gradient variance during early training. In contrast, random sampling does not provide these benefits. Therefore, the gain of the Dense Batching stage arises from the dense batching mechanism itself rather than from acting as a generic warm-up.

---

### Author Response · Authors · 2026-06-24
**General Summary**

Dear All Reviewers and Action Editor,

We sincerely appreciate all the reviewers for their thoughtful comments and suggestions on our paper. We are deeply grateful for the Action Editor's effort and time to manage our submission.

We are very glad to see that all reviewers think that our proposed method achieves great empirical gains and our methodology is conceptually simple, well-motivated, and computationally lightweight with low engineering overhead. Furthermore, reviewer gEo9 finds our work practically useful and relevant to efficient LLM pretraining, and reviewer rUFL acknowledges the Token Utilization Rate (TUR) formulation as a useful conceptual perspective for understanding pretraining efficiency.

We have tried our best to address the reviewers' comments and concerns in individual responses to each reviewer with comprehensive experimental justifications and theoretical clarifications. The reviews allowed us to substantially improve our draft. **Please note that in the revised version of our manuscript, all modified text and additions have been explicitly highlighted in light green for better readability.** The main contents added and updated in the revised version are summarized below:

**From Reviewer rUFL**
- Conduct comparison experiments against modern packing-based pretraining implementations (see Appendix A.3.1)
- Conduct additional long-context experiments (see Appendix A.3.4)
- Conduct an ablation experiment to disentangle the contribution of the Dense Batching stage from a generic warm-up effect (see Figure 5(b) and Section 4.3)

**From Reviewer gEo9**
* Provide end-to-end runtime analysis (see Table 1, Figure 4(a), and Appendix A.3.4)
* Clarify data separation (see Section 3.2).
* Qualify the theoretical claims (see Abstract and Section 3.3)
* Moderate claims about downstream improvements (see Section 4.2)
* Clarify the distinctions between DeBLAS and prior fixed length-curriculum methods and modern packing-based training implementations (see Appendix A.3.1 and Appendix B).
* Conduct sensitivity analysis for calibration-specific hyperparameters (see Appendix A.2.2)
* Conduct analysis for stability of learned length-bin sampling probabilities (see Section 4.3)
* Revise Broader Impact Statement

**From Reviewer VFTT**
* Revise the claim on packing (see Section 2 and Appendix A.3.1)
* Revise the claim on foundational language knowledge internalization (see Abstract)
* Revise the claim in Broader Impact Statement
* Provide hyperparameter selection guidelines (see Appendix A.2.2).
* Redesign Figure 3(c) (see Section 3.1 and Appendix A.3.2).
* Provide analysis for memory-related metrics (see Section 4.3).
* Provide more detailed table caption (see Table 4)

Thanks once again for all the reviewers' valuable time and efforts. Your insightful comments have been invaluable in enhancing our manuscript and have motivated us to further advance this research.

Best regards,

Authors of #8323 TMLR submission